# Robust single-nucleus RNA sequencing reveals depot-specific cell population dynamics in adipose tissue remodeling during obesity

Jisun So[1], Olivia Strobel[1], Jamie Wann[1], Kyungchan Kim[1], Avishek Paul[1], Dominic J Acri[2], Luke C Dabin[2], Jungsu Kim[2], Gang Peng[3], Hyun Cheol Roh[1]*

[1]Department of Biochemistry and Molecular Biology, Indiana University School of Medicine, Indianapolis, United States; [2]Stark Neurosciences Research Institute, Indiana University School of Medicine, Indianapolis, United States; [3]Department of Medical and Molecular Genetics, Indiana University School of Medicine, Indianapolis, United States

*For correspondence:
hyunroh@iu.edu

## eLife Assessment

So et al. present an optimized protocol for single-nuclei RNA sequencing of adipose tissue in mice, ensuring better RNA quality and nuclei integrity. The authors use this protocol to explore the cellular landscape in both lean and diet-induced obese mice, identifying a dysfunctional hypertrophic adipocyte subpopulation linked to obesity. The data analyses are **solid**, and the findings are supported by the evidence presented. This study provides **valuable** information for the field of adipose tissue biology and will be particularly helpful for researchers using single-nuclei transcriptomics in various tissues.

**Abstract** Single-nucleus RNA sequencing (snRNA-seq), an alternative to single-cell RNA sequencing (scRNA-seq), encounters technical challenges in obtaining high-quality nuclei and RNA, persistently hindering its applications. Here, we present a robust technique for isolating nuclei across various tissue types, remarkably enhancing snRNA-seq data quality. Employing this approach, we comprehensively characterize the depot-dependent cellular dynamics of various cell types underlying mouse adipose tissue remodeling during obesity. By integrating bulk nuclear RNA-seq from adipocyte nuclei of different sizes, we identify distinct adipocyte subpopulations categorized by size and functionality. These subpopulations follow two divergent trajectories, adaptive and pathological, with their prevalence varying by depot. Specifically, we identify a key molecular feature of dysfunctional hypertrophic adipocytes, a global shutdown in gene expression, along with elevated stress and inflammatory responses. Furthermore, our differential gene expression analysis reveals distinct contributions of adipocyte subpopulations to the overall pathophysiology of adipose tissue. Our study establishes a robust snRNA-seq method, providing novel insights into the biological processes involved in adipose tissue remodeling during obesity, with broader applicability across diverse biological systems.

## Introduction

Single-cell RNA sequencing (scRNA-seq) has revolutionized various fields of biology by enabling high-resolution profiling of single-cell heterogeneity within complex tissues (*Tang et al., 2009*). It has not only facilitated the identification of new cell types and subtypes (*Shaffer et al., 2017*), but also uncovered alterations in cellular composition and gene expression across diverse biological contexts during development, disease progression and responses to environmental stimuli (*Shalek et al., 2013*; *Trapnell et al., 2014*; *Petropoulos et al., 2016*). However, experimental methods for scRNA-seq still present limitations and challenges. First, these methods require the use of fresh tissues, which can be difficult to obtain and thus lead to logistic complexities (*Alles et al., 2017*; *Attar et al., 2018*; *Wang et al., 2018*; *Lafzi et al., 2018*). Even when fresh tissues are available, the process of obtaining single-cell suspension through enzymatic digestion and mechanical dissociation can introduce unwanted artifacts. Different types of cells can be digested to varying degrees and rates, resulting in biased cell recovery. Also, the tissue dissociation process can cause cellular stress, leading to reduced cell viability and altered transcriptional profiles. Lastly, certain cell types are unsuitable for scRNA-seq due to their unique cellular properties (*Grindberg et al., 2013*; *Nguyen et al., 2018*). Adipocytes are one example; their large size and excess lipid content render them incompatible with the microfluidic platforms used in the majority of scRNA-seq technologies.

To overcome these issues, single-nucleus RNA sequencing (snRNA-seq) has been developed as an alternative strategy. While generating transcriptional profiles with high similarity to those by scRNA-seq (*Grindberg et al., 2013*; *Habib et al., 2017*), snRNA-seq offers advantages. First, this method utilizes nuclei that are promptly released from tissues during homogenization, obviating the need for an extended digestion process and thereby minimizing cell type biases and artifactual transcriptional alteration. Second, its compatibility with cryopreserved samples allows for greater flexibility in experimental designs and enables the utilization of samples stored in biobanks. Furthermore, snRNA-seq can be employed to study cell types that are less amenable to scRNA-seq, such as adipocytes and cardiomyocytes, as isolated nuclei no longer retain limiting cellular characteristics and can be loaded on to various platforms. Nevertheless, snRNA-seq also poses challenges to be considered. Once released from cells, nuclei are highly fragile and susceptible to mechanical stress, resulting in leakage or aggregation. Moreover, their enclosed RNA becomes increasingly vulnerable to RNases, leading to rapid RNA degradation. Nuclei isolated from in vivo tissues are typically more delicate than those from cells cultured in vitro, with potential variation depending on cell type. Hence, meticulous preparation of high-quality nuclei is crucial for successful snRNA-seq experiments.

Adipose tissue plays a pivotal role in energy metabolism, serving as a dynamic energy reservoir within the body. It exhibits remarkable plasticity in response to nutritional fluctuations (*Rosen and Spiegelman, 2014*). In obesity, adipose tissue undergoes a significant remodeling process, expanding substantially to enhance its storage capacity. This process is orchestrated through coordinated actions among diverse cell types, including adipocytes, preadipocytes, endothelial cells, and various immune cells (*Corvera, 2021*). However, prolonged obesity leads to pathological changes and dysfunction in adipose tissue, characterized by disrupted lipid metabolism, impaired insulin signaling, inflammation, and cell death. These changes are critical precursors to the onset of type 2 diabetes and other metabolic diseases (*Longo et al., 2019*). The distribution of body fat in obesity is closely correlated with metabolic health. The expansion of visceral fat depots increases risks for metabolic disorders, whereas subcutaneous fat depots pose lesser risks or can even offer protection (*Vishvanath and Gupta, 2019*). This is likely due to varying responses of different adipose tissue depots to obesity (*Pellegrinelli et al., 2016*). In obesity, visceral adipose tissues display enlarged hypertrophic adipocytes, along with hypoxia, elevated fibrosis and inflammation from increased immune cell infiltration. In contrast, subcutaneous adipose tissues have smaller adipocytes with lower levels of inflammation and hypoxia (*Longo et al., 2019*). scRNA-seq has been used to characterize the dynamics of cell populations underlying adipose tissue remodeling, with a focus on the cell types within stromal vascular fractions, particularly preadipocytes (*Schwalie et al., 2018*; *Burl et al., 2018*; *Hepler et al., 2018*; *Merrick et al., 2019*). Yet adipocytes have been excluded from these studies due to technical limitations. More recently, snRNA-seq has been employed, revealing evidence of the cellular heterogeneity within adipocyte populations (*Sárvári et al., 2021*; *Emont et al., 2022*; *Holman et al., 2023*). Despite this progress, acquiring high-quality snRNA-seq data from adipose tissues remains highly challenging. Technical difficulties have posed a major hurdle to achieving

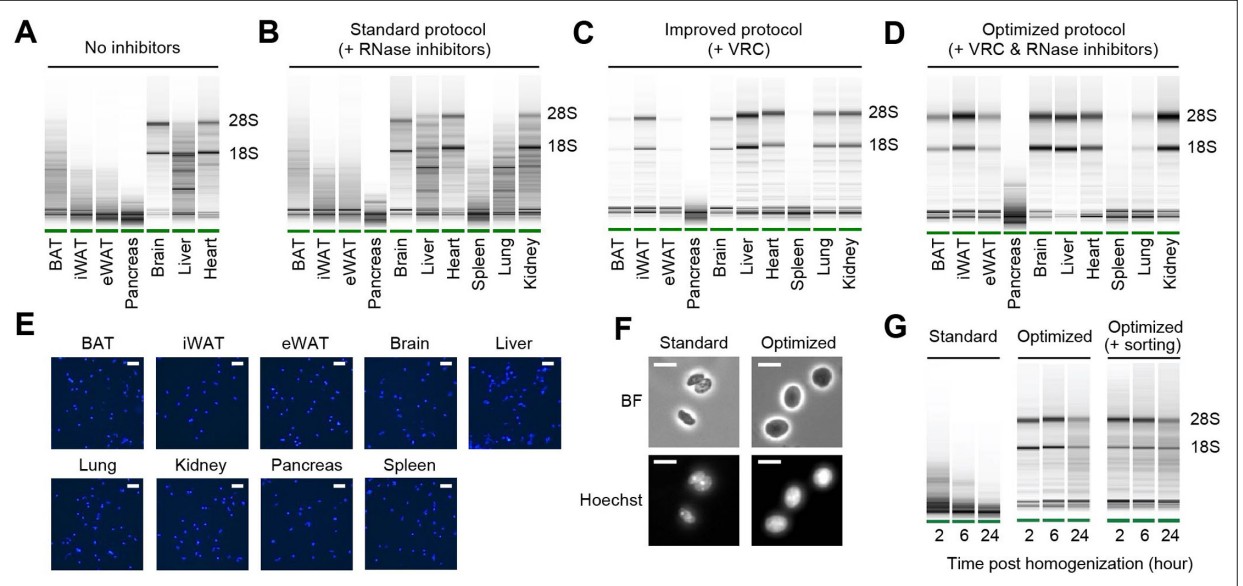

**Figure 1.** RNA quality and nucleus integrity are preserved by robust nucleus isolation protocols. (**A–D**) Analysis of RNA quality in isolated nuclei using Agilent Bioanalyzer. The presence of distinct 18S and 28S bands indicates high RNA quality. Nuclei were isolated from indicated tissue types using different protocols: (**A**) No RNase inhibitor included, (**B**) standard protocol with RNase inhibitors (0.4 U/µl), (**C**) improved protocol with vanadyl ribonucleoside complex (VRC; 10 mM), and (**D**) optimized protocol using both VRC and RNase inhibitors. (**E**) Immunofluorescence microscopy images of nucleus suspensions stained with Hoechst (1 µM) from indicated tissue types. Scale bars: 20 µm. (**F**) Brightfield (BF) and immunofluorescence microscopy images of Hoechst-stained nuclei isolated from epididymal white adipose tissue (eWAT) using the standard and optimized protocols. Scale bars: 10 µm. (**G**) RNA quality assessed by Agilent Bioanalyzer of isolated nuclei from eWAT stored at 4°C for the indicated duration post homogenization.

accuracy, sensitivity, and efficacy in understanding adipose tissue remodeling, especially regarding mature adipocyte populations during obesity.

Here, we present a robust method for isolating nuclei for snRNA-seq, which enhances the quality of nuclear RNA and preserves nucleus integrity across various tissue types. Using this method, we have generated snRNA-seq data of superior quality compared to existing datasets, offering novel and precise insights into the specific changes within cell populations across different fat depots in response to obesity. By integrating bulk nuclear RNA-seq with adipocyte nuclei of different sizes, we have identified multiple subpopulations of adipocytes primarily categorized by size and functionality, which are influenced by obesity. Specifically, we have characterized dysfunctional hypertrophic adipocytes prevalent in visceral adipose tissues during obesity, exhibiting hallmark features of cellular stress and inflammation. All these adipocyte subpopulations contributed differentially to a range of biological pathways involved in the pathophysiology of adipose tissue during obesity.

## Results
### Robust nucleus isolation protocols protect RNA quality and nucleus integrity

To systemically address the issue of nuclear RNA degradation, we first assessed RNA quality in isolated nuclei from a range of mouse tissues. During nucleus isolation, no RNase inhibitors were added to evaluate intrinsic RNase activity in each tissue type. We observed a wide spectrum of RNA quality among the tissues examined (*Figure 1A*). The brain exhibited highly intact RNA, followed by the heart showing relatively good RNA quality. The liver displayed moderate RNA degradation. Among the adipose tissues, including brown adipose tissue, subcutaneous inguinal white adipose tissue (iWAT), and visceral epididymal white adipose tissue (eWAT), significant RNA degradation was detected, with eWAT showing the highest level of degradation. RNA was completely degraded in the pancreas (*Figure 1A*). These results indicate significant variation in the quality of RNA extracted from nuclei depending on tissue types. To limit RNA degradation, we incorporated recombinant RNase inhibitors during nucleus isolation and examined additional tissue types, including the spleen, the lung, and the

kidney. Surprisingly, the addition of recombinant RNase inhibitors at a concentration of 0.4 U/µl had no discernible effect on RNA quality of any of those tissues (*Figure 1B*). Even when the concentration of RNase inhibitors was increased to 4 U/µl, it failed to yield any substantial enhancement in RNA quality (data not shown). Therefore, we screened various conditions and molecules known for their ability to inhibit RNase activity. Among these, the vanadyl ribonucleoside complex (VRC) (*Lienhard et al., 1972*) was identified as having a significant impact on RNA quality. High-quality RNA with minimal degradation was obtained in all the tested tissues except for the pancreas and its connecting tissue, the spleen (*Figure 1C*). Finally, we optimized the protocol by combining VRC with recombinant RNase inhibitors for further improvements, albeit to a modest extent (*Figure 1D*). In addition to RNA quality, achieving a single-nucleus suspension is an important requirement for snRNA-seq sample preparation. Using VRC, the optimized protocol successfully retrieved a well-dispersed nucleus population without clumps in all tissue types (*Figure 1E*). Further analysis of nucleus morphology at a higher resolution revealed a smoother membrane surface and larger nucleus size when using the optimized protocol, as compared to the standard protocol (*Figure 1F*). Consistent with the preservative function of VRC proven in maintaining tissue morphology (*Shieh et al., 2018*), these findings show that our nucleus isolation protocol effectively maintains nucleus structure without shrinkage. Given that current snRNA-seq methods require prompt sample processing due to the rapid degradation of RNA in isolated nuclei, we further tested the temporal stability of RNA in nuclei isolated using our optimized protocol. After isolating nuclei from eWAT, we stored them at 4°C for different durations (2, 6, and 24 hr) before RNA extraction and quality analysis. In the standard protocol, nuclear RNA exhibited severe degradation within 2 hr post-homogenization, worsening over 24 hr. In contrast, nuclear RNA isolated using the optimized protocol remained intact at 2 and 6 hr, maintaining good quality even after 24 hr (*Figure 1G*). To determine whether RNA could remain intact with additional handling procedures, we conducted the same time-course RNA extraction with nuclei sorted through flow cytometry. The RNA remained stable, comparable to that of unsorted nuclei (*Figure 1G*). Taken together, our optimized protocol, employing both VRC and recombinant RNase inhibitors, yields high-quality nuclei while preserving RNA integrity and nuclear structure for extended procedures up to 24 hr.

## The optimized protocol improves the data quality of snRNA-seq

Considering the previous report on the inhibitory activity of VRC on reverse transcriptases (*Shieh et al., 2018*), we investigated whether the inclusion of VRC could confound the accurate representation of RNA expression profiles within nuclei. By utilizing the Nuclear tagging and Translating Ribosome Affinity Purification (NuTRAP) mice designed to label nuclei from specific cell types (*Roh et al., 2017*), we sorted nuclei from adipocytes and non-adipocytes to conduct a comparative analysis of their gene expression profiles through cDNA synthesis followed by quantitative real-time PCR (qRT-PCR). We observed strong PCR amplification and highly specific expression of adipocyte markers in adipocyte nuclei and immune/endothelial cell markers in non-adipocyte nuclei (*Figure 2-figure supplement 1A*). These results indicate that the use of VRC during nucleus isolation does not pose an issue in gene expression profiling.

Subsequently, we applied the optimized nucleus isolation protocol for snRNA-seq of mouse adipose tissues using the 10x Genomics Chromium Single Cell 3′ Gene Expression system. We employed a two-step tissue homogenization process involving pulverization and Dounce homogenizers to ensure complete liberation of nuclei (*Figure 2A*). Additionally, we included a fluorescence-activated nucleus sorting (FANS) step in the workflow to eliminate tissue debris and ambient RNA (*Figure 2—figure supplement 1B*). From eWAT and iWAT samples collected from mice fed either a chow or high fat diet (HFD), we generated high-quality cDNA with long fragment sizes and constructed robust sequencing libraries (*Figure 2—figure supplement 1C, D*). To evaluate our data quality, we benchmarked them against two published snRNA-seq datasets from mouse adipose tissues (*Sárvári et al., 2021*; *Emont et al., 2022*), based on the quality metrics determined by the standard Cell Ranger pipeline. With comparable sequencing depths (*Figure 2B*), our datasets displayed significantly higher counts of unique molecular identifiers (UMIs) and genes per nucleus, exceeding counts from the other datasets by almost twofold (*Figure 2C, D*). Also, the proportions of mitochondrial reads in our datasets represented less than 1% (mean 0.37%), which was significantly lower than in the other datasets (mean 38.7 and 2.31%; *Figure 2E*). Importantly, our method produced substantially higher fractions of reads in

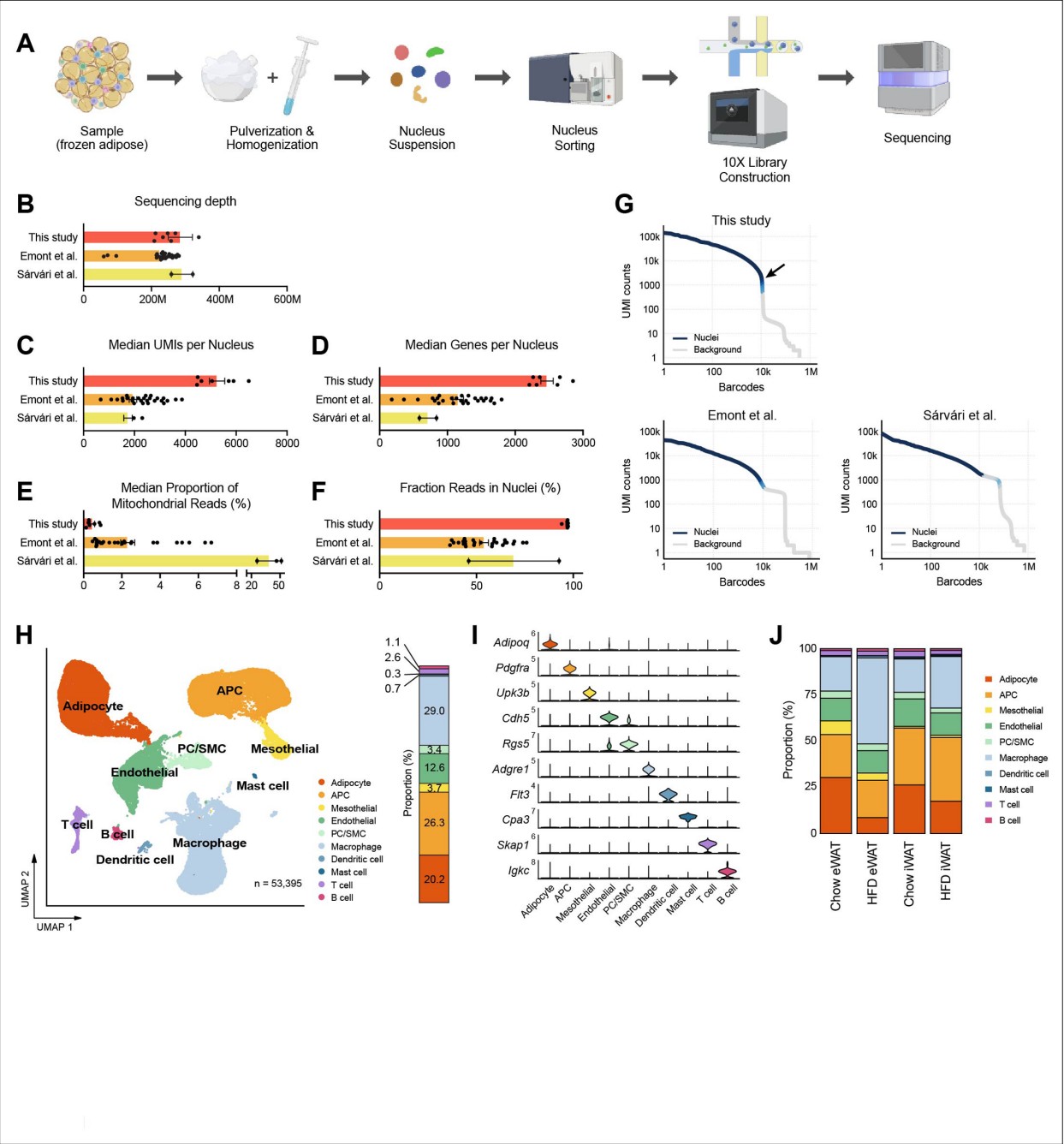

**Figure 2.** A single-nucleus atlas of white adipose tissue from lean and obese mouse generated by the optimized single-nucleus RNA sequencing (snRNA-seq) protocol. (**A**) Schematic of the workflow for the optimized protocol for snRNA-seq of mouse frozen tissue samples. (**B–F**) Comparison of the quality of snRNA-seq datasets included in this study (*n* = 7) with those from previous studies (***Emont et al., 2022***, *n* = 27; ***Sárvári et al., 2021***, *n* = 2) using mouse adipose tissue. Bars indicate mean ± SEM, with each dot corresponding to an individual dataset. (**B**) Sequencing depth, (**C**) median numbers of unique molecular identifiers (UMIs) detected per nucleus, (**D**) median number of genes detected per nucleus, (**E**) median proportion of reads originating from mitochondrial genes, and (**F**) fraction of reads in nucleus-associated barcodes. (**G**) Barcode rank plots show the ranks of barcodes based on UMI counts, generated from this study and previous studies (***Sárvári et al., 2021***; ***Emont et al., 2022***). Darker blue shading indicates a higher proportion of nucleus versus background barcodes. The arrow indicates the steep cliff between nucleus versus background barcodes. (**H**) Uniform manifold approximation and projection (UMAP) of all 53,395 single nuclei isolated from epididymal white adipose tissue (eWAT) and inguinal white adipose tissue (iWAT) of lean and obese mice. The UMAP-based clustering was performed with a resolution of 0.6, and clusters were annotated as specific cell types with the indicated colors based on their marker genes. The bar on right side displays the overall relative proportions of individual cell types across the samples. (**I**) Violin plots showing the expression levels of selected marker genes representing each cell type in mouse adipose tissue.

*Figure 2 continued on next page*

*Figure 2 continued*

(**J**) Bar graphs showing the relative proportions of individual cell types in adipose tissue by fat depot and diet. APC, adipocyte progenitor cells; PC, pericytes; SMC, smooth muscle cells.

The online version of this article includes the following figure supplement(s) for figure 2:

**Figure supplement 1.** Precise and robust gene expression profiling by our optimized single-nucleus RNA sequencing (snRNA-seq) protocol.

**Figure supplement 2.** Profile of our single-nucleus atlas encompassing diverse cell populations.

nuclei (*Figure 2F*), suggesting low ambient RNA contamination. This was consistent with the barcode rank plot for our data, which exhibited a sudden drop of UMI counts (*Figure 2G*), indicating a clear distinction between intact nuclei and background barcodes during cell calling. Analysis using Cell-Bender (*Fleming et al., 2023*) also contrasted the minimal number of cells in the narrow transition zone in our datasets with the greater number of cells within the broader transition zone observed in the published samples (*Figure 2—figure supplement 1E*). These results indicate that our method reduces the contribution of ambient RNA to the expression profiles of single cells. Collectively, these findings suggest that our optimized protocol produces high-quality snRNA-seq data that outperforms other existing methods.

## A single-nucleus atlas of different adipose tissues from lean and obese mice

To characterize the cellular dynamics within mouse adipose tissues during obesity, we integrated data from both eWAT and iWAT of chow- or HFD-fed wild-type male mice. After filtering nuclei based on the detected numbers of UMIs and genes, and subsequently removing doublets and ambient RNA, a total of 53,395 cells/nuclei were identified, distributed across 10 distinct cell types (*Figure 2H*). The major cell types present in adipose tissues included adipocytes, adipocyte progenitor cells (APC), macrophages/monocytes, and endothelial cells. Additionally, we identified less abundant cell types, such as pericytes (PC)/smooth muscle cells (SMC), mesothelial cells, dendritic cells, mast cells, T cells, and B cells. We determined distinct marker gene sets for each cell type, including both established and novel markers (*Figure 2I*, *Figure 2—figure supplement 2A*). The proportions of these cell types in our data largely matched those reported in previous studies, except for the notable improvement in recovering vascular cells (endothelial cells and PC/SMC) (*Figure 2—figure supplement 2B*). This is likely due to our robust two-step tissue homogenization, which effectively releases nuclei from vasculature that would otherwise be mechanically resistant.

For a more robust comparison, we integrated our data with the previous study by *Emont et al., 2022* and re-clustered all the cells together. The results consistently showed the similar overall cellular composition with a substantially higher proportion of vascular cells in our dataset (*Figure 2—figure supplement 2C*). We found that the UMI and genes per nucleus were higher in every cell type in our data except for adipocytes (*Figure 2—figure supplement 2D*), which included a dysfunctional subtype with low transcript abundance to be discussed below. These findings suggest that the new protocol provides significant advantages in generating a more accurate representation of each cell type for tissue cell atlases.

The cellular composition altered substantially in response to HFD feeding, with variations depending on the fat depot (*Figure 2J*). HFD resulted in a relative reduction in adipocytes and an increase in macrophage/monocyte fractions in both eWAT and iWAT, with more prominent changes in eWAT (*Figure 2J*, *Figure 2—figure supplement 2E*). APC fractions were smaller in eWAT compared to iWAT under chow diet, with marginal effects under HFD. The population size of endothelial cells did not show notable differences between the conditions (*Figure 2—figure supplement 2E*). Most of the minor cell populations exhibited minimal differences across the conditions, except for dendritic cells which increased in eWAT in response to HFD, and mast cells which were more abundant in iWAT than eWAT (*Figure 2—figure supplement 2E*).

## Obesity induces the transition of APC from an early state to a committed state in specifically eWAT

We next performed subclustering of each cell type found in mouse adipose tissues to gain insight into the cellular states and dynamics of cell subtypes during obesity, focusing on their unique gene

expression profiles and proportional changes during obesity in different depots. Analysis of APCs revealed four different subpopulations (*Figure 3A, B*), which generally aligned with prior studies. APC1 and APC2 corresponded to 'adipocyte progenitors', characterized by high expression of both *Dpp4* and *Pi16* (*Merrick et al., 2019Merrick et al., 2019*), while displaying distinct gene expression profiles from each other (*Figure 3—figure supplement 1A*). Notably, APC2 exhibited higher expression of fibrotic genes such as *Fn1* and *Loxl1* (*Figure 3—figure supplement 1B*), resembling 'fibro-inflammatory progenitors' (*Hepler et al., 2018*). APC3 was identified as 'adipose regulatory cells', expressing *F3* (encoding CD142) and showed strong expression of secretory signaling proteins (*Figure 3A*, *Figure 3—figure supplement 1C*), supporting its paracrine regulatory role (*Schwalie et al., 2018*). APC4, the largest subpopulation, represented 'committed preadipocytes', expressing lipid metabolism-related genes such as *Lpl*, *Hsd11b1*, and *Fabp4* (*Figure 3A*, *Figure 3—figure supplement 1D*). The distribution of APC subpopulations differed between adipose depots, with eWAT displaying a higher fraction of APC3 and APC4, while iWAT showed a larger fraction of APC1 (*Figure 3C*). Specifically in eWAT, there was an expansion of the APC4 proportion alongside a reduction in APC1 and APC2 populations (*Figure 3C*) during obesity, suggesting distinct tissue expansion mechanisms in each adipose tissue depot. eWAT appears to promote de novo adipogenesis through conversion of adipocyte progenitors to committed preadipocytes, as demonstrated in a prior study (*Wang et al., 2013*).

## Adipose tissue immune cells acquire pro-inflammatory phenotypes predominantly in eWAT during obesity

We identified seven subpopulations of macrophages and monocytes based on distinct marker expression profiles (*Figure 3D, E*, *Figure 3—figure supplement 2A*). The two largest populations, MC1 and MC2, expressed *Clec10a* (encoding CD301) and *Mrc1* (encoding CD206), indicative of alternatively activated M2 macrophages (*Figure 3—figure supplement 2A*). However, MC2 stood out for its enrichment of genes associated with antigen presentation and integrin binding, including major histocompatibility complex markers such as *H2-Eb1* and *H2-Aa* (*Figure 3—figure supplement 2B, C*). These MC1 and MC2 cells corresponded perivascular-like and non-perivascular-like macrophages, respectively, as described by *Sárvári et al., 2021*. MC3 represented 'lipid-associated macrophages (LAMs)', characterized by high expression of lipid metabolism genes (*Figure 3—figure supplement 2D*). We found two minor macrophage subpopulations, MC6 and MC7, which resembled previously described regulatory macrophages (RegM) and collagen-expressing macrophages (CEM), respectively (*Figure 3D*, *Figure 3—figure supplement 2A*). Their prevalence showed depot-dependent differences, with eWAT demonstrating a higher abundance of RegM and iWAT containing more CEM (*Figure 3F*). MC4 and MC5, accounting for approximately 20% of the myelocyte populations, were identified as classical monocytes and non-classical monocytes, respectively (*Figure 3D*, *Figure 3—figure supplement 2A*). In response to HFD feeding, only eWAT exhibited a marked relative reduction in MC1 and MC2 populations with a substantial increase in MC3 (LAMs) (*Figure 3F*).

Lymphocytes were clustered into four T cell subpopulations – regulatory (Treg), CD4+, CD8+, and natural killer T cells – along with two B cell subpopulations (*Figure 3G, H*, *Figure 3—figure supplement 2E*). The distribution of T cell subpopulations differed between adipose depots: CD4+ and CD8+ T cells were more abundant in iWAT, whereas Treg were more prevalent in eWAT (*Figure 3I*). Conversely, B cells exhibited relatively high homogeneity in proportions between fat depots under chow conditions (*Figure 3G, I*). The most remarkable change in lymphocytes during HFD feeding was the emergence of a small B cell subpopulation, BC2, observed in eWAT (*Figure 3G*), which highly expressed genes involved in B cell proliferation, such as *Prkar2b* and *Mcc* (*Figure 3—figure supplement 2F*). Collectively, our high-quality data enabled accurate and detailed characterization of diverse immune cell populations in adipose tissue, revealing their dynamic changes during obesity in a depot-dependent manner.

## Obesity-induced changes in vascular cell subpopulations represent depot-dependent dynamics of adipose tissue remodeling

The higher recovery of vascular cells in our snRNA-seq data provided comprehensive insights into the vasculature of adipose tissues. We identified nine subpopulations of vascular cells (*Figure 3J, K*), grouped into three major types: blood endothelial cells (blood EC), mural cells, and lymphatic

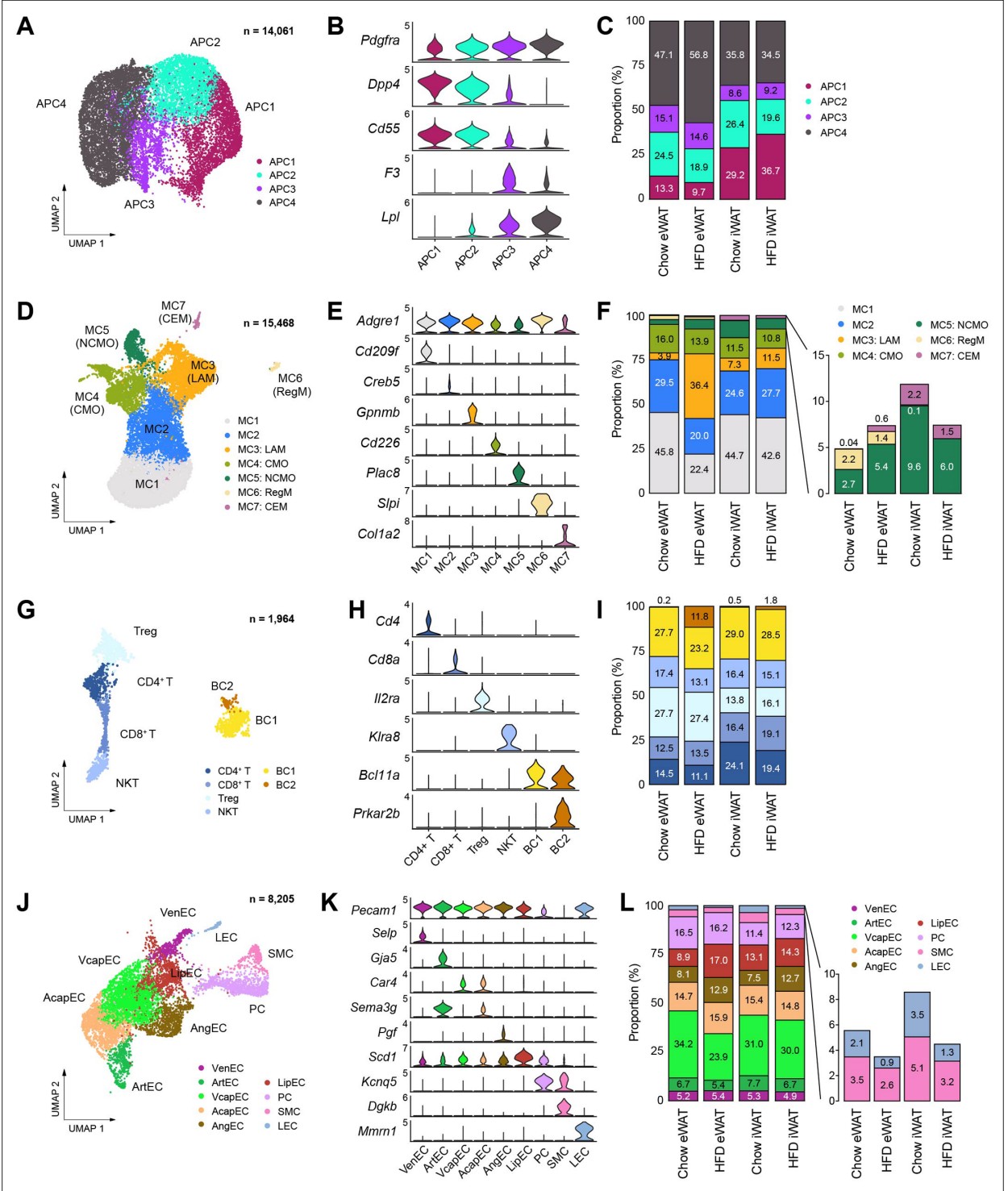

**Figure 3.** Subpopulations of cell types within the stromal vascular fraction of mouse adipose tissue that vary across fat depot and diet. (**A–C**) Identification of adipocyte progenitor cell (APC) subpopulations. (**A**) Uniform manifold approximation and projection (UMAP), (**B**) marker genes, and (**C**) the relative proportions of APC subpopulations (*n* = 14,061). (**D–F**) Identification of macrophage/monocyte subpopulations. (**D**) UMAP projection, (**E**) marker genes, and (**F**) the relative proportions of macrophage/monocyte subpopulations (*n* = 15,468). (**G–I**) Identification of lymphocyte subpopulations. (**G**) UMAP projection, (**H**) marker genes, and (**I**) the relative proportions of lymphocyte subpopulations (*n* = 1964). (**J–L**) Identification of vascular cell subpopulations. (**J**) UMAP projection, (**K**) marker genes, and (**L**) the relative proportions of vascular cell subpopulations (*n* = 8205). CEM, collagen-expressing macrophages; CMO, classical monocytes; LAM, lipid-laden macrophage; LEC, lymphatic endothelial cells; LipEC, lipid-associated

*Figure 3 continued on next page*

*Figure 3 continued*

endothelial cells; NCMO, non-classical monocytes; NKT, natural killer T cells; PC, pericytes; RegM, regulatory macrophages; SMC, smooth muscle cells; Treg, regulatory T cells; VcapEC, venous capillary endothelial cells; VenEC, venous endothelial cells.

The online version of this article includes the following figure supplement(s) for figure 3:

**Figure supplement 1.** Gene expression profiles of adipocyte progenitor cell (APC) subpopulations indicating distinct functional states.

**Figure supplement 2.** Characterization of diverse immune cell subpopulations.

**Figure supplement 3.** Characterization of diverse vascular cell subpopulations.

endothelial cells (LEC) (*Figure 3—figure supplement 3A*). Blood EC included venous (VenEC), arterial (ArtEC), and capillary endothelial cells (CapEC). CapEC was further divided into arterial capillary endothelial cells (AcapEC) and venous capillary endothelial cells (VcapEC) (*Figure 3—figure supplement 3B*), with the latter being the largest vascular cell type in adipose tissues (*Figure 3L*). We additionally recognized angiogenic endothelial cells (AngEC), comprising migratory tip cells expressing extracellular matrix (ECM) genes, and proliferative stalk cells with elevated expression of cell cycle genes (*Figure 3—figure supplement 3B*). Furthermore, we discovered an endothelial cell type characterized by high expression of lipid metabolism genes (LipEC) (*Figure 3—figure supplement 3B*). Following HFD feeding, a relative expansion of AngEC and LipEC was observed in both fat depots (*Figure 3L*), suggesting increased vascularization and lipid transport during obesity-induced adipose tissue expansion. Intriguingly, we noted a drastic decrease in the proportion of the VcapEC population specifically within eWAT during HFD feeding (*Figure 3L*), indicating impaired metabolite transport into the venous circulation, which may potentially contribute to the pathology of visceral adipose tissue.

In mural cells, PC and SMC were identified (*Figure 3—figure supplement 3C*), with the PC population notably larger than that of SMC (*Figure 3L*), consistent with their respective association with capillaries and larger vessels (*Rhodin, 1968*; *Yamazaki and Mukouyama, 2018*). Lastly, we identified LEC (*Figure 3—figure supplement 3D*), comprising 1–3% of the vascular cell population. Interestingly, the fraction of LECs diminished following HFD feeding in both eWAT and iWAT (*Figure 3L*). The HFD-induced reduction of LEC, along with the aforementioned decline in VcapEC, implies that metabolic drainage into lymphatic vessels and the venous circulation could be a key pathological feature of visceral adipose tissues during obesity.

## Adipocyte subpopulations embody diverse functional and cellular states influenced by obesity

snRNA-seq has recently begun to reveal the heterogeneity of adipocytes within white adipose tissue (*Sárvári et al., 2021*; *Emont et al., 2022*; *Holman et al., 2023*). In our dataset, we identified six subpopulations of adipocytes, characterized by a gradient of unique gene expression signatures among them rather than distinct cell types (*Figure 4A, B*). Ad1–Ad3 represented a profile of mature adipocytes with high expression of insulin signaling genes and lipid metabolism genes, with Ad2 demonstrating particularly robust expression of lipid metabolism genes (*Figure 4—figure supplement 1A*). Ad3, along with Ad4, also showed elevated expression of genes associated with cell junction, ECM organization and calcium homeostasis (*Figure 4—figure supplement 1B*). Ad6 was distinguished by its pronounced expression of genes involved in unfolded protein binding, oxidative stress, cytoskeleton, antigen presentation, and adipokines such as *Lep*, *Retn*, and *Rbp4* (*Figure 4—figure supplement 1C*). In response to HFD feeding, the proportions of Ad1 and Ad2 were significantly reduced in both eWAT and iWAT, while Ad3 showed a relative increase in iWAT. Notably, the Ad6 proportion exhibited a substantial increase in eWAT and a moderate rise in iWAT (*Figure 4C*, *Figure 4—figure supplement 1D*). These findings indicate that the identified adipocyte subpopulations may represent diverse functional and cellular states influenced by nutritional conditions.

Adipocytes undergo hypertrophy, an increase in cell size, during obesity, accompanied by significant cytoskeletal rearrangement (*Roh et al., 2020*). Thus, we questioned whether the classification of adipocyte subpopulations aligns with the spectrum of adipocytes during HFD-induced expansion, particularly regarding their size. To address this question, we sought to compare gene expression profiles between large and small adipocytes. Interestingly, we observed enlarged nuclei in hypertrophic adipocytes isolated from HFD-fed mice (*Figure 4D*), which correlated positively with adipocyte

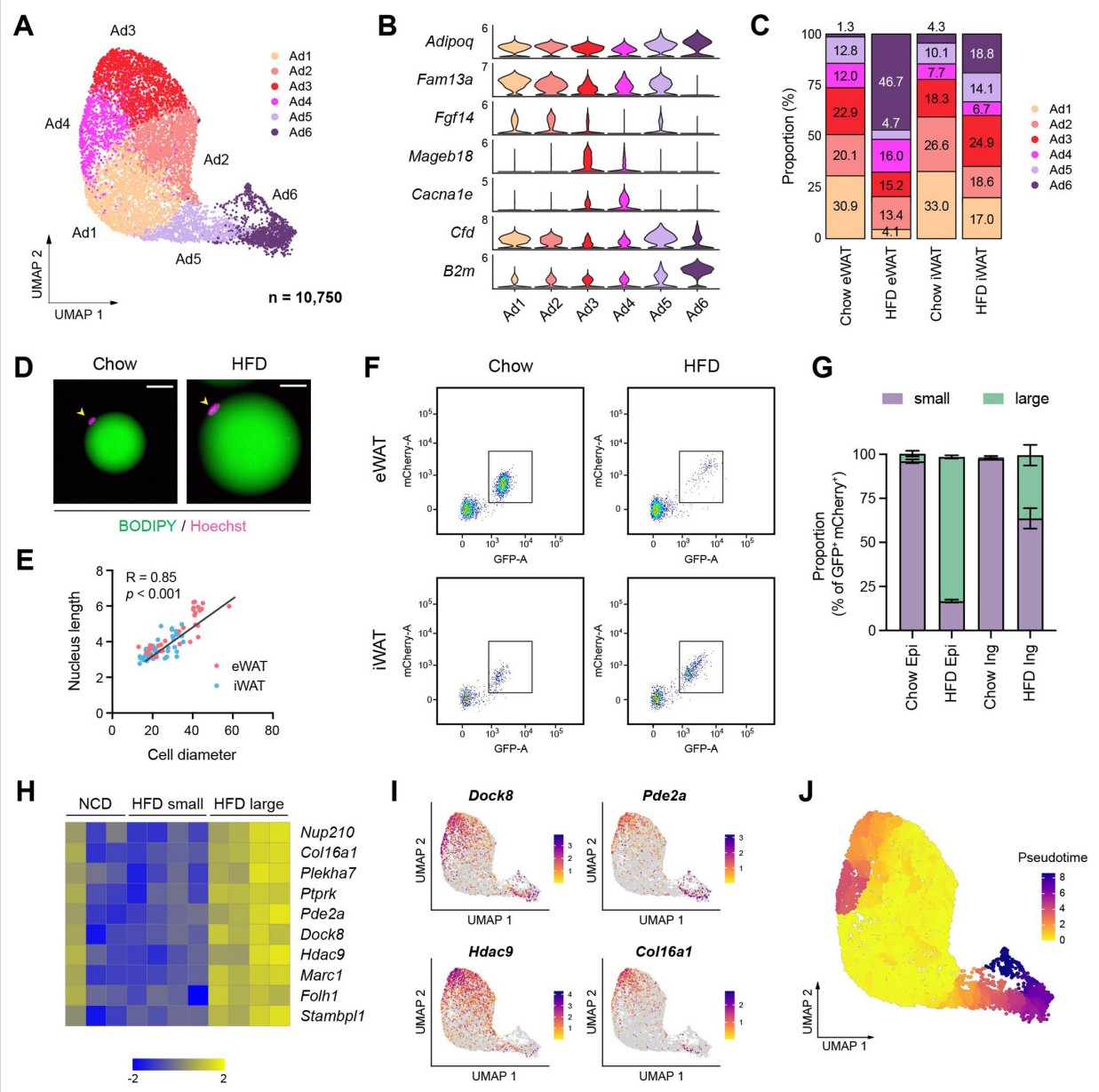

**Figure 4.** Distinct functional and cellular states of adipocytes affected by obesity. (**A–C**) Identification of adipocyte subpopulations. (**A**) Uniform manifold approximation and projection (UMAP), (**B**) marker genes, and (**C**) the relative proportions of adipocyte subpopulations (*n* = 10,750). (**D**) Immunofluorescence microscopy images of BODIPY- and Hoechst-stained adipocytes isolated from epididymal white adipose tissue (eWAT) in chow- and high fat diet (HFD)-fed mice. Scale bars: 20 µm. (**E**) Scatterplot illustrating the correlation between adipocyte size, measured by diameter, and nucleus size, measure by length, in eWAT (red) and inguinal white adipose tissue (iWAT, blue). (**F**) Scatterplot illustrating the size distribution of mCherry- and GFP-labeled adipocyte nuclei from NuTRAP mice across different fat depots and diets. (**G**) The relative proportions of small and large nuclei within mCherry- and GFP-labeled adipocyte nuclei across different fat depots and diets. Data are presented as mean ± SEM (*n* = 4). See *Figure 4—figure supplement 1D* for gating criteria. (**H**) Heatmap showing Z-scored expression of hypertrophy signature genes in iWAT adipocyte nuclei from chow-fed mice and in small and large adipocyte nuclei from HFD-fed mice. The signature genes were identified based on enrichment in large versus small adipocyte nuclei in HFD-fed mice. (**I**) UMAP visualization of select hypertrophy signature genes in adipocytes. (**J**) Representation of pseudotime within adipocytes, as identified by Monocle3.

The online version of this article includes the following figure supplement(s) for figure 4:

**Figure supplement 1.** Identification of functional attributes of adipocyte subpopulations.

cell size in both fat depots (*Figure 4E*). Profiling adipocyte nuclei labeled with GFP and mCherry from NuTRAP mice fed on chow or HFD showed a strong positive correlation between nuclear GFP intensity and adipocyte nucleus size (*Figure 4—figure supplement 1E*). Based on these findings, we analyzed the size distribution of adipocyte nuclei in each fat depot under chow or HFD conditions using flow cytometry. eWAT and iWAT from chow-fed mice primarily displayed smaller adipocyte nuclei. In HFD-fed mice, eWAT predominantly contained larger adipocyte nuclei, while iWAT showed a heterogeneous size range from small to large nuclei (*Figure 4F, G*). Next, we sorted and separately collected adipocyte nuclei by size from the same iWAT of HFD-fed mice for RNA-seq analysis (*Figure 4—figure supplement 1F*) to define the genuine gene expression signature associated with cell/nucleus size, minimizing potential confounding factors such as diet or depot dependency. Differential gene expression analysis followed by gene ontology analysis revealed that large adipocytes promote processes involved in insulin response, vascularization and DNA repair, while inhibiting processes related to cell migration, metabolism and the cytoskeleton (*Figure 4—figure supplement 1G, H*). We also identified adipocyte hypertrophy signature genes enriched in large adipocyte nuclei compared to smaller adipocyte nuclei (*Figure 4H*). They also exhibited higher overall expression levels in adipocyte nuclei in iWAT from HFD-fed mice compared to those from chow-fed mice (*Figure 4H*). When overlaying the expression of these hypertrophy signature genes across adipocyte subpopulations, we observed high expression in Ad3–Ad4 and Ad5–Ad6 (*Figure 4I*), suggesting that they may represent hypertrophic adipocytes. However, certain genes differentiated between Ad3–Ad4 and Ad5–Ad6. For example, *Hdac9* was more enriched in Ad3, while *Pde2a* was in more enriched in Ad5–Ad6 (*Figure 4I*). Importantly, there were marked distinctions in the expression of insulin signaling- and lipid metabolism-related genes between Ad3 and Ad6, with Ad6 having substantially lower gene

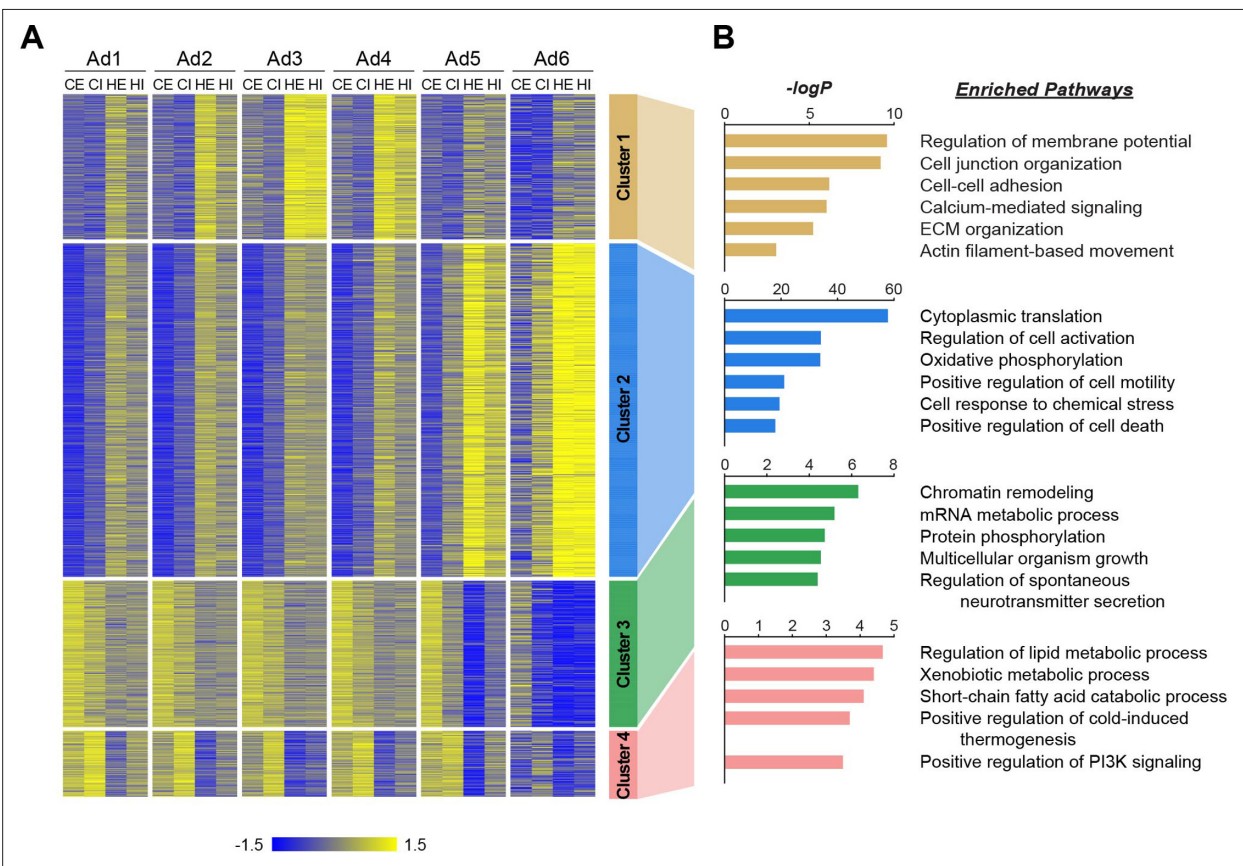

**Figure 5.** Distinct biological pathways enriched in different adipocyte subpopulations during obesity. (**A**) Heatmap illustrating *Z*-scored expression of high fat diet (HFD)-induced genes in either epididymal white adipose tissue (eWAT) or inguinal white adipose tissue (iWAT) within each adipocyte subpopulation. Those genes are classified into four different clusters by *K*-means clustering based on their expression patterns across the samples. (**B**) Top pathways enriched within each cluster of genes with their corresponding −logP values, as determined by Metascape. CE, chow eWAT; CI, chow iWAT; HE, HFD eWAT; HI, HFD iWAT.

expression levels (*Figure 4—figure supplement 1A*). This implies that Ad3 represents 'metabolically healthy hypertrophic (MHH)' adipocytes, while Ad6 represents 'metabolically unhealthy hypertrophic (MUH)' adipocytes.

To further discern the obesity-driven dynamics of adipocyte subpopulations, we performed pseudotime trajectory analysis, starting with the Ad1–Ad2 subpopulations that were prevalent under chow conditions (*Figure 4J*). Two divergent trajectories were revealed: One trajectory moves toward Ad3 and Ad4, which may depict an 'adaptive healthy transition'. Conversely, the alternate trajectory progresses toward Ad5 and ultimately to Ad6, indicating a potential 'maladaptive dysfunctional transition'. In addition, the farthest position of Ad6 along the pseudotime scale (*Figure 4J*) indicates prolonged pathological conditions. Importantly, eWAT and iWAT displayed contrasting preferences in adipocyte trajectories during HFD feeding: the dysfunctional trajectory leading to Ad6 was dominant in eWAT while the adaptive trajectory toward Ad3 remained more prevalent in iWAT (*Figure 4C*). This supports the depot-dependent dynamics of adipocytes that contribute to distinct metabolic outcomes during obesity.

## Adipocyte subpopulations differentially contribute to tissue-level changes during obesity

We next aimed to evaluate how individual adipocyte subpopulations contribute to tissue-level alterations during obesity. First, we identified genes that were differentially expressed in response to HFD within each subpopulation in eWAT or iWAT. Subsequently, we classified these genes into four clusters based on their pattern of HFD-induced changes across adipocyte subpopulations (*Figure 5A, Supplementary file 1*). Gene cluster 1 represented a group of genes generally upregulated by HFD across all subpopulations, with the most significant induction observed in Ad3, followed by Ad4 and Ad2 to a lesser extent. The genes associated with cell junctions, ECM and the cytoskeleton included in this cluster (*Figure 5B*) may describe biological responses during HFD-induced adipose tissue remodeling, which were featured in the aforementioned adaptive transition, among Ad2-Ad4 subpopulations. On the other hand, Gene clusters 2 and 3 comprised genes up- and down-regulated, respectively, with changes most pronounced in Ad6 and to a lesser extent in Ad5, during HFD feeding. The induction of genes in cluster 2, as well as the repression of genes in cluster 3, reflect stress response, inflammation, and cell death in adipose tissues, along with impaired machinery regulating gene expression (e.g., chromatin remodeling and mRNA metabolic process) during HFD-induced adipose pathology (*Figure 5B*). Genes in cluster 4, commonly down-regulated by HFD across all adipocyte subpopulations, had enriched metabolic processes such as lipid metabolism, thermogenesis, and insulin phosphatidylinositol 3-kinase (PI3K) signaling (*Figure 5B*), indicating a general decline in metabolic activity and insulin sensitivity across the entire adipocyte population rather than specific subpopulations. Collectively, these results suggest that tissue-level biological changes in adipose tissues during obesity result from a combination of alterations within specific subpopulations and throughout the entire adipocyte population.

## Ad6 may represent stressed and dying adipocytes with transcriptional shutdown

Our snRNA-seq data unveiled a previously unrecognized adipocyte subpopulation, Ad6, predominantly present in eWAT during HFD feeding. This subpopulation represented a dysfunctional state of adipocytes, characterized by elevated expression of stress response and inflammation genes, coupled with the loss of metabolic genes. Using the identified profile as a basis, we sought to confirm the presence of the Ad6 adipocyte subpopulation at the tissue level through adipocyte or whole-mount immunostaining. We first visualized reactive oxygen species (ROS) within adipocytes isolated from adipose tissues of chow- or HFD-fed mice using the chemical ROS reporter H2DCFDA. Under HFD conditions, we observed markedly higher H2DCFDA fluorescence intensity in eWAT and a modest increase in iWAT, resulting in a significantly larger fraction of H2DCFDA-positive adipocytes, especially in eWAT (*Figure 6A, B*). This provides evidence of Ad6 adipocytes experiencing oxidative stress. Next, we inspected Ad6 adipocytes by employing whole-mount immunostaining of adipose tissues for beta-2 microglobulin (B2M), a marker highly enriched in this subpopulation (*Figure 4—figure supplement 1C*).

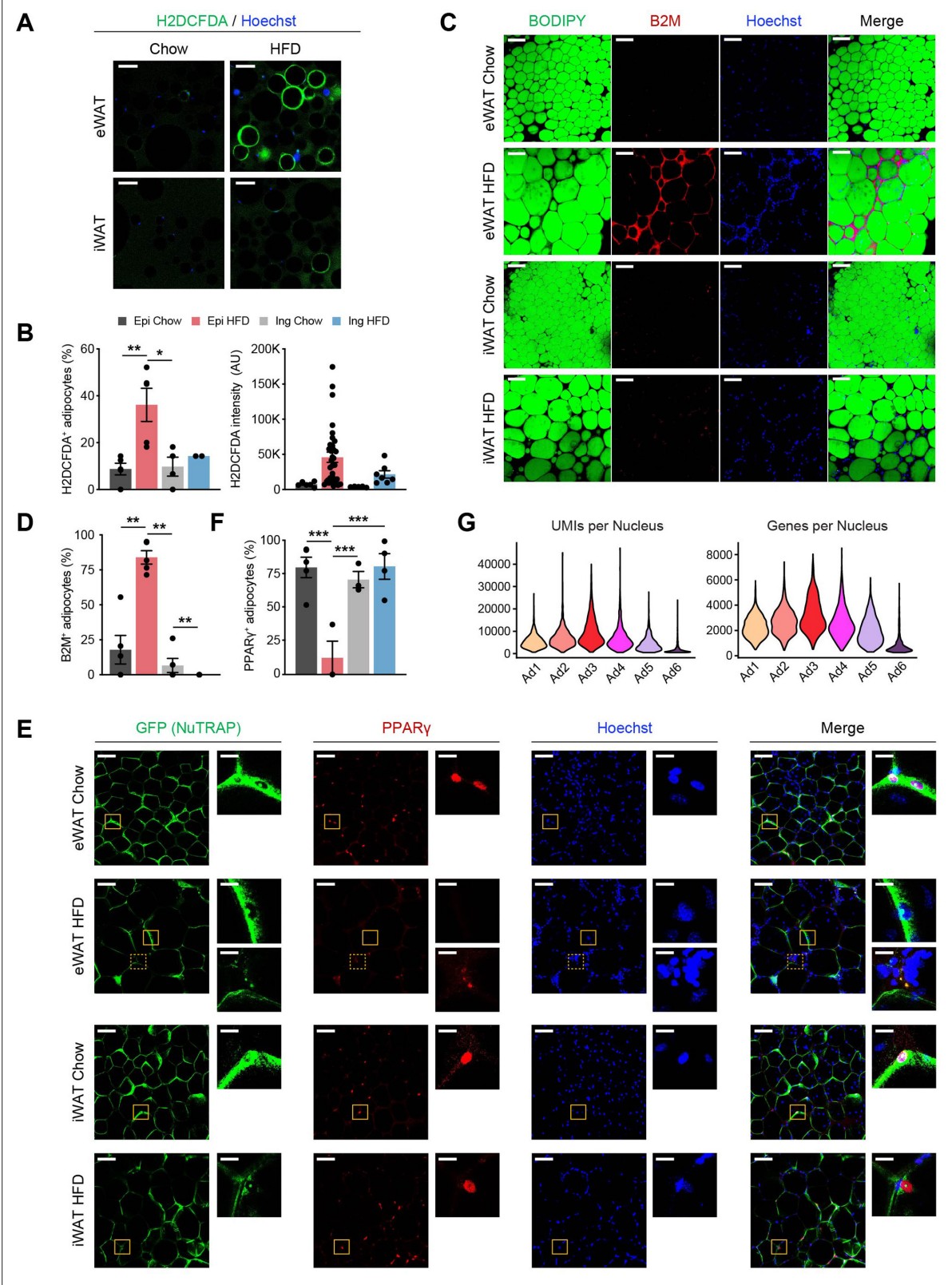

**Figure 6.** The Ad6 subpopulation represents the pathological state of adipocytes during obesity. (**A**) Representative immunofluorescence microscopy images of adipocytes isolated from epididymal white adipose tissue (eWAT) and inguinal white adipose tissue (iWAT) of chow- and high fat diet (HFD)-fed mice stained with Hoechst and the chemical reactive oxygen species (ROS) reporter, H2DCFDA. Scale bars: 100 μm. (**B**) Quantification of H2DCFDA-positive adipocyte fraction (left) and intensity (right) per image in eWAT (*n* = 5) and iWAT (*n* = 2–4) from chow- and HFD-fed mice. (**C**) Representative

*Figure 6 continued on next page*

*Figure 6 continued*

images of whole-mount adipose tissue stained with BODIPY, B2M, and Hoechst by fat depot and diet. Scale bars: 50 μm. (**D**) Quantification of B2M-positive adipocyte fraction per image by fat depot and diet (*n* = 5, each). (**E**) Representative images of whole-mount eWAT and iWAT from chow- and HFD-fed NuTRAP mice stained with GFP, PPARγ, and Hoechst. The insets display select areas of GFP-labeled adipocyte nuclei. Note: The inset with the dotted line on eWAT HFD represents autofluorescence, not PPARγ. Scale bars: 100 and 20 μm (insets). (**F**) Quantification of PPARγ-positive adipocyte nucleus fraction among GFP-labeled adipocyte nuclei in eWAT (*n* = 3–5) and iWAT (*n* = 3–4) from chow- and HFD-fed mice. (**G**) Numbers of unique molecular identifiers (UMIs; left) and genes (right) detected per nucleus in each adipocyte subpopulation. Statistical analysis was performed using one-way ANOVA with Tukey post hoc test. *p < 0.05, **p < 0.01, ***p < 0.001.

The online version of this article includes the following figure supplement(s) for figure 6:

**Figure supplement 1.** Comparison of adipocyte subpopulations identified in this study versus *Emont et al., 2022*.

---

B2M was strongly detected in eWAT and slightly in iWAT after HFD feeding. B2M was particularly found within crown-like structures (CLS), as well as in the cytoplasm of adipocytes neighboring these CLS (*Figure 6C, D*). As dead adipocytes are detected within CLS (*Murano et al., 2008*), this result suggests Ad6 as stressed and dying cells due to inflammation, both within and outside the CLS. Another hallmark of Ad6 adipocytes was the depletion of genes crucial for lipid metabolism, including peroxisome proliferator-activated receptor γ (PPARγ) (*Figure 4—figure supplement 1A*). To mitigate the potential for artifacts derived from ambient RNA, we directly visualized PPARγ proteins within adipocytes in adipose tissues through whole-mount immunostaining utilizing NuTRAP mice. While most GFP-labeled adipocyte nuclei robustly expressed PPARγ proteins under typical conditions, a significant proportion of adipocyte nuclei from eWAT in HFD-fed mice lacked PPARγ proteins (*Figure 6E, F*), confirming the presence of PPARγ-deficient Ad6 adipocytes in eWAT during obesity. We did not detect significant expression of APC markers such as *Pdgfra* in Ad6 (*Figure 4—figure supplement 1I*), indicating that Ad6 does not represent dedifferentiating adipocytes. Instead, the substantially fewer transcripts, represented by UMIs or genes, detected in the Ad6 subpopulation (*Figure 6G*), suggests Ad6 adipocytes undergo a global loss of gene expression. In the integration of our data with previous datasets, we observed a comparable adipocyte population identified in the previous study (mAd3 in *Emont et al., 2022Emont et al., 2022*; *Figure 6—figure supplement 1A*), with similar marker gene expression (*Figure 6—figure supplement 1B*) and lower transcript abundance (*Figure 6—figure supplement 1C*). However, the population size of mAd3 was much smaller than Ad6 in our data and did not show consistent population changes during obesity (*Figure 6—figure supplement 1D*). This discrepancy may be due to different technical robustness; the dysfunctional cellular state of this population, with severely compromised RNA contents, may have made it difficult to accurately capture using standard protocols in the previous study, while our protocol enabled robust and precise detection. Taken together, Ad6 represents a subpopulation of stressed and dying adipocytes with transcriptional shutdown that arise during obesity.

## Discussion

snRNA-seq has emerged as an alternative approach to overcome the limitations of scRNA-seq. Nevertheless, technical challenges in obtaining high-quality nuclei and RNA have remained as significant hurdles in the standard protocol. In this study, we present a robust protocol for nucleus isolation that effectively addresses these issues across various tissue types. To demonstrate the efficacy of our method, we applied it to characterize one of the most challenging tissue types to investigate, adipose tissue. Our snRNA-seq data demonstrate its superior quality compared to previously published datasets, as well as provides novel insights into the biological processes underlying mouse adipose tissue remodeling during obesity.

We observed that significant variations in RNase abundance among various tissue types affect the quality of RNA obtained from isolated nuclei using standard protocols. Notably, brain tissue, which has been primarily used in prior snRNA-seq studies, exhibited the lowest RNase activity. This implies that the standard nucleus isolation protocols employed in those studies may not readily translate to other tissues, particularly those with high RNase activities. Recombinant RNase inhibitor proteins are commonly used to control RNA quality during nucleus isolation. However, these inhibitors marginally improved RNA quality, likely due to their specific targeting of certain RNase types, such as RNase A, 1, and T1. Instead, through experiments, we discovered that VRC, a potent inhibitor capable of

targeting a wide range of RNases, significantly enhanced RNA quality in isolated nuclei. VRC functions as a transition state analog, binding to the active sites of RNases and inhibiting their enzymatic activity (*Gray, 1974*; *Berger and Birkenmeier, 1979*). Despite its demonstrated versatility in RNA experiments (*Gray, 1974*; *Berger and Birkenmeier, 1979*; *Egberts et al., 1977*), the application of VRC in gene expression studies has been limited due to its inhibitory effects on other RNA enzymes, including reverse transcriptase (*Berger and Birkenmeier, 1979*). Our method addressed this issue by reducing VRC from nucleus isolation buffer prior to loading onto microfluidic devices, as confirmed by the robust qRT-PCR and snRNA-seq results we obtained. Recently, we encountered a few protocols that have been introduced to enhance RNA quality by employing VRC or acidic buffers during nucleus isolation for snRNA-seq, particularly for kidney (*Soccio et al., 2014*) and pancreas (*Tosti et al., 2021*) samples. However, their applicability to other tissue types remains untested. In contrast, our optimized protocol successfully preserved not only nuclear RNA quality in various tissue types, but also the integrity of the nuclear structure. We noted that VRC has preservative effects on isolated nuclei, resulting in smoother and more intact membranes in larger nuclei when compared to standard protocols not using VRC. This observation aligns with a recent study that demonstrated the preservative properties of VRC, protecting tissue morphology, as well as RNA, protein, and genomic DNA quality (*Shieh et al., 2018*). More importantly, we demonstrated the capability of our protocol to retain nuclear RNA quality for up to 24 hr post-tissue processing, even after undergoing additional steps such as FANS. This advancement holds the potential to greatly improve experimental logistics. Considering that snRNA-seq experiments typically require completion of nucleus isolation and loading onto microfluidic devices (e.g., 10x Genomics Chromium controller) within a narrow time window on the same day, our protocol offers greater flexibility for these critical steps, allowing them to be performed on different days if necessary. This adaptability has the potential to unlock a wide array of applications for snRNA-seq techniques. In the clinical setting, for example, this will enable patient biopsies to be collected for diagnostic tests and then be analyzed via snRNA-seq at a different facility.

Our single-nucleus atlas of mouse white adipose tissues generally aligns with previous studies while offering deeper insights into depot specificity and cellular composition dynamics during obesity. An important advancement in our dataset was the improved recovery of vascular cell populations, which were underrepresented in prior studies, likely due to our protocol's effective tissue homogenization and the high quality of RNA preserved in isolated nuclei. This enabled precise characterization of diverse endothelial and related vascular cell types. We observed a marked decrease in the proportion of capillary endothelial cells in eWAT specifically during HFD feeding, consistent with the previously documented capillary rarefaction and elevated hypoxia associated with adipose tissue expansion (*Rosen and Spiegelman, 2014*; *Pasarica et al., 2009*). We further noticed that the VcapEC population, which is responsible for transporting metabolites into the venous circulation, was primarily affected. Additionally, we observed an HFD-induced decline in the LEC population, which plays a role in removing fluid and macromolecules from the interstitial space. These findings suggest that defective drainage of metabolites and molecules into the venous circulation may be another key feature of dysfunctional adipose tissue in obesity, warranting further investigation.

Another noteworthy discovery from our snRNA-seq data was gaining deep insights into the spectrum of adipocyte subpopulations and their dynamics during obesity. Based on our observation of a strong correlation between adipocyte cell size and nucleus size, we developed a novel approach to profile the size distribution of adipocyte subpopulations using nuclei as surrogates, which yielded intriguing findings. Adipocyte cell size appeared small and relatively uniform under chow conditions in both eWAT and iWAT. During HFD feeding, adipocytes notably enlarged, particularly in eWAT, whereas iWAT exhibited a broader distribution with a mixture of small and large adipocytes. Therefore, by isolating adipocyte nuclei of various sizes from iWAT, we identified gene signatures that determine adipocyte hypertrophy. Overlaying this data with snRNA-seq data revealed that cell size is a key determinant of adipocyte clustering in mice. This finding contrasts with the heterogeneity of human adipocytes, where insulin sensitivity is shown to be a primary factor in adipocyte classification rather than cell size, as demonstrated by spatial transcriptomics mapping (*Massier et al., 2023*). Interestingly, ex vivo differentiated human adipocytes derived from various donors displayed distinct cellular phenotypes, consistent with the heterogeneity in adipocyte subpopulations identified by snRNA-seq (*Emont et al., 2022*). This suggests that the significant genetic variation among humans, potentially serving as a major driver of adipocyte clustering, overrides the size-dependent differences

found in inbred mouse strains with homogeneous genetic backgrounds. Nevertheless, our findings integrate various characteristics of adipocytes reported from diverse biological contexts, broadening our understanding of distinct adipocyte subpopulations. For instance, despite both Ad1 and Ad2 subpopulations having small cell size and abundance under chow conditions, they exhibited distinct expression levels of lipid metabolism genes. This finding may be consistent with previous studies demonstrating mosaic lipid transport activity among small white adipocytes from *Rhesus Macaques* monkeys (*Varlamov et al., 2015*). Also, the unique enrichment of genes involved in calcium transport and ossification in hypertrophic Ad4 adipocytes may account for the frequent occurrence of calcium deposits observed in obese adipocytes (*Giordano et al., 2013*).

We further noted two distinct types of hypertrophic adipocytes developed by HFD feeding: MHH adipocytes (Ad3) abundant in iWAT, and MUH adipocytes (Ad6) prevalent in eWAT. We not only validated the presence of MUH adipocytes, primarily around CLS in obese eWAT with oxidative stress and inflammation markers, but also discovered their potential association with prolonged obesity through pseudotime analysis. Interestingly, MUH adipocytes showed deficient expression of the master transcriptional regulator PPARγ as well as a substantial decline in metabolic gene expression. This aligns with our earlier findings, which demonstrated the failure of adipocytes to maintain their identity during obesity (*Roh et al., 2020*). However, MUH adipocytes did not show significant expression of marker genes for APC, indicating that they are not in a state of dedifferentiation. The significant decrease in transcript counts in Ad6 suggests that these cells may represent stressed or dying cells, akin to the 'stressed lipid-scavenging' adipocytes described by *Sárvári et al., 2021*. Our data integration analysis found that Ad6 adipocytes are comparable to mAd3 adipocytes identified in the study by *Emont et al., 2022*. However, we observed significant discrepancies in the size and changes of this population during obesity, likely due to differences in technical robustness. We did not find any human adipocyte subclusters that clearly resembled our mouse Ad6 adipocytes. Notably, human adipocyte heterogeneity does not correspond well to that of mouse adipocytes (*Massier et al., 2023*). Furthermore, human adipocyte data show poor reproducibility between different studies (*Massier et al., 2023*). Interestingly, this inconsistency is unique to adipocytes, as other cell types in adipose tissues display consistent sub cell types across species and studies (*Massier et al., 2023*). Our findings indicate that adipocytes may exhibit a unique pathological cellular state with significantly reduced RNA content, which may contribute to the poor consistency in adipocyte heterogeneity in prior studies with suboptimal RNA quality. Therefore, using a robust method to effectively preserve RNA quality may be critical for accurately characterizing adipocyte populations, especially in disease states. It remains to be tested in the future studies whether our snRNA-seq protocol can identify consistent heterogeneity in adipocyte populations across different species, studies, and individual human subjects. Also, examining short- and mid-term obesity conditions could help clarify cell population dynamics and identify early markers involved in the transition from healthy to pathological adipose tissue.

Lastly, we elucidated the contribution of distinct adipocyte subpopulations to various established signatures of obese adipose tissue. For instance, Ad2–Ad4 adipocytes appeared to undergo tissue remodeling for enhanced adipose functionality, such as storage capacity. In contrast, the traditional features of dysfunctional adipose tissue, including inflammation, cell death, and stress response, were predominantly derived from Ad5–Ad6 adipocytes. They were also responsible for chromatin alterations and defective gene expression. On the other hand, the core functions of adipocytes, including lipid metabolism and insulin signaling, were generally impaired in all adipocyte subpopulations. The distinctions between obesity-induced features occurring in the entire adipocyte population versus specific adipocyte subpopulations may explain the outstanding effectiveness of PPARγ agonists, such as thiazolidinediones, in improving adipose function and glucose homeostasis (*Soccio et al., 2014*), surpassing the effects of other anti-inflammatory or anti-stress reagents (*Kraakman et al., 2015*; *Chen et al., 2016*; *da Cruz Nascimento et al., 2022*). Our discovery emphasizes the need to tailor therapeutic approaches based on the specific biological processes to be targeted to improve adipose tissue function during obesity.

In conclusion, this study introduces a robust nucleus isolation method for snRNA-seq, leading to enhanced data quality. This advancement has enabled the discovery of novel insights into the dynamics of adipose tissue remodeling during obesity. In addition, this method holds great promise for widespread applications across various research fields and clinical diagnoses, involving numerous types of biological samples.

## Materials and methods

### Animal studies

All animal experiments were performed according to a protocol approved by the Indiana University School of Medicine (IUSM) Institutional Animal Care and Use Committee (IACUC).

The mice were housed under a 12-hr light–dark cycle at 22°C with free access to food and water.

To label adipocyte nuclei, NuTRAP mice (Jackson Laboratory, 029899) were crossed with Adipoq-Cre mice (Jackson Laboratory, 010803). In experiments assessing nuclear RNA quality, male C57BL/6J mice at 8–10 weeks of age were used for tissue collection. To generate diet-induced obesity models, male mice were subjected to either a HFD (60% kcal from fat, Research Diets D12492i) or a standard chow diet (control) from 8 weeks of age for a duration of 10 weeks. Upon dissection, tissues were immediately snap-frozen in liquid nitrogen. During the dissection of eWAT, we carefully removed epididymal ducts from the adipose tissues, minimizing the risk of contamination. For iWAT, the lymph nodes were removed to avoid over-representation of lymphocytes.

### Nucleus isolation

For each experiment, the nucleus preparation buffer (NPB) was freshly prepared at room temperature with the following components: 10 mM 4-(2-hydroxyethyl)-1-piperazineethanesulfonic acid (HEPES, pH 7.5), 1.5 mM $MgCl_2$, 10 mM KCl, 250 mM sucrose, 0.1% NP-40, and 1 mM DL-dithiothreitol (DTT). Next, VRC (New England Biolabs) was added to the NPB at a concentration of 10 mM. The mixture was vigorously vortexed until the VRC was completely dissolved in the NPB. The NPB with VRC was then chilled on ice, and 0.4 U/µl of NxGen RNase inhibitor (Biosearch Technologies) and 1× Halt protease inhibitor (Thermo Fisher) were added. Once the NPB was ready, frozen tissue samples were pulverized using mortars and pestles in liquid nitrogen. The resulting tissue powders were immediately transferred to and homogenized in chilled Douncers placed on ice, using 5–7 ml of the NPB. The tissue homogenates were filtered through 100 µm cell strainers and centrifuged at 200 × $g$ for 10 min at 4°C. After removing the supernatant, the nucleus pellet was resuspended in 5 ml of PBS-N-VRC, which is calcium and magnesium-free phosphate-buffered saline (PBS) with 0.1% NP-40 and 5 mM VRC. Hoechst 33342 (Thermo Fisher) was added to the tissue homogenates at a concentration of 1 µg/ml to label nuclei. The mixture was then centrifuged at 200 × $g$ for 5 min at 4°C. Subsequently, the nucleus pellets were resuspended in PBS-N-VRC buffer and filtered through 40 µm cell strainers. The protocol for nucleus isolation is currently under the provisional patent application process.

### Nuclear RNA extraction for quality analysis

The nucleus resuspension in PBS-N-VRC was centrifuged at 200 × $g$ for 10 min at 4°C, and the supernatant was discarded. To reduce VRC amount, the nucleus pellet was resuspended in PBS-N (PBS with 0.1% NP-40), and then centrifuged again at 200 × $g$ for 10 min. After removing the supernatant, the nucleus pellet was subjected to RNA extraction using TRIzol as per the manufacturer's instructions. Briefly, the nucleus pellet was thoroughly resuspended in TRIzol (Invitrogen) by vigorous vortexing and subsequently mixed with chloroform. Following centrifugation at 13,000 × $g$ for 10 min at 4°C, the aqueous phase was collected and combined with isopropanol and 1.5 µg of GlycoBlue coprecipitant (Invitrogen). After a 10-min incubation at room temperature, the mixture was centrifuged at 13,000 × $g$ for 10 min at 4°C. The RNA pellet was washed with 75% ethanol and then dried. Subsequently, the pellet was resuspended in nuclease-free water, and the RNA concentration was measured using a Qubit Fluorometer with the Qubit RNA High Sensitivity Assay Kit (Invitrogen). The quality of the RNA was assessed using a high sensitivity electrophoresis system, the Agilent Bioanalyzer. Since the automatically generated RNA integrity number provided by the system did not accurately reflect the quality of nuclear RNA due to the presence of a substantial amount of small RNA molecules, we opted for a qualitative analysis of the RNA quality. For time-course experiments to assess RNA quality over time, the nuclear suspension in PBS-N-VRC was kept at 4°C for the indicated duration prior to RNA extraction.

For visual inspection of nucleus quality, the nucleus resuspension was mounted on hemacytometer and visualized using a Zeiss Observer.Z1 fluorescence microscope using bright field and DAPI filters.

## snRNA-seq with nucleus sorting

The nucleus resuspension in PBS-N-VRC was subjected to nucleus sorting using a BD FACSAria Fusion system. We set gating based on forward scatter, side scatter, and Hoechst fluorescence to eliminate debris and nucleus doublets/multiplets. Sorted nuclei were collected in PBS-N-VRC, then centrifuged at 200 × g for 10 min at 4°C using a swing bucket rotator. After removing the supernatant, the nucleus pellet was resuspended in PBS-N, filtered through 20 µm cell strainers, and centrifuged at 200 × g for 5 min at 4°C in a swing bucket rotator. Subsequent to removing the supernatant, the nucleus pellet was resuspended in PBS-N in volumes aimed at achieving a nucleus density of approximately 1000 nuclei/µl. The nuclei in the resuspension were counted using a hematocytometer or automated cell counters. Following the counting, 16,000 nuclei (with a target of 10,000) were loaded onto the 10x Genomics Chromium controller. Libraries were prepared using the Chromium Single Cell 3′ Reagent Kit v3.3 (10x Genomics), indexed using sample barcodes, and subsequently sequenced on an Illumina NovaSeq 6000 sequencer with paired-end dual indexing.

## Bulk nuclear RNA-seq with nucleus sorting

For bulk nuclear RNA-seq, nuclei were sorted based on the aforementioned gating criteria following isolation. Sorted nuclei were washed in PBS-N as for snRNA-seq but skipping additional filtration. Nuclear RNA was extraction using TRIzol as stated above. Additionally, the extracted RNA was purified by on-column DNA digestion using TURBO DNase kit (Invitrogen). For library construction, 20–50 ng of purified RNA was processed by NEBNext rRNA Depletion Kit (New England BioLabs, Inc) for removing ribosomal RNA, Next the RNA was converted to cDNA using Maxima Reverse Transcriptase (Thermo Fisher Scientific) and NEBNext mRNA Second Strand Synthesis kit. Following purification based on size selection using AMPure XP beads (Beckman Coulter), sequencing libraries were generated through tagmentation using Nextera XT DNA Library Preparation kit and subsequent PCR amplification. The quantity and quality of the resulting libraries were assessed using Qubit and Agilent Bioanalyzer, respectively. Finally, the libraries were sequenced on an Illumina NextSeq500.

## H2DCFDA staining with isolated adipocytes

Fresh adipose tissues were minced and digested in PBS containing collagenase D (1.5 U/ml) and dispase II (2.4 U/ml) with shaking for 30–40 min at 37°C. After digestion, the tissue homogenates were filtered through 250 µm cell strainers and left to settle for 10 min. Floating adipocytes were collected, washed in PBS containing 1% fat-free bovine serum albumin (BSA), and incubated with H2DCFDA (Invitrogen, 20 uM) and Hoechst 33342 (Thermo Fisher) for 30 min in the dark at room temperature. The cells were washed in PBS containing 0.05% Tween20 and 0.1% fat-free BSA twice, placed on a chamber made on slides using 90% glycerol, and covered with coverslips. The slides were visualized using a Leica SP8 confocal microscope.

## Whole-mount immunostaining

Fresh adipose tissues were fixed in 10% buffered formalin phosphate for 1 day and then washed in PBS. Subsequently, the fixed tissues were permeabilized in PBS containing 1% Triton X-100 for 1 hr, following by blocking in PBS containing 5% BSA and 0.05% Tween20, for another hour. The tissues were incubated with (1) anti-B2M (Proteintech) or (2) anti-GFP (Novus Biologicals) and anti-PPARg (Cell Signaling Technology) at 1/200 dilution for 1 day. After washes in PBS/0.05% Tween20, the tissues were further incubated with Alexa Fluor-conjugated antibodies at 1/100 dilution for 1 hr. For B2M staining, the tissues were additionally incubated with BODIPY 493/503 (Invitrogen) for 30 min. After washed twice in PBS/0.05% Tween20 including once with Hoechst, the tissues were mounted on 35 mm glass bottom dish using glycerol mounting buffer and covered with coverslips. The mounted tissues were visualized using a Leica SP8 confocal microscope.

## Quantitative real-time PCR

Total RNA was extracted from sorted nuclei using TRIzol, and the extracted RNA was then converted into cDNA using the High-Capacity cDNA Reverse Transcription Kit from Applied Biosystems. qRT-PCR was conducted using the SYBR Green PCR Master Mix on a QuantStudio5 system. The fold change was determined using the ΔΔCT method by comparing target gene expression with a reference gene 36B4 (*Rplp0*). The primers used for qRT-PCR are provided in *Supplementary file 2*.

## Bioinformatics analysis

### Sequencing, alignment, and quality control

Data were processed using Cell Ranger (10x Genomics, v7.1.0) to demultiplex raw sequencing reads to FASTQ format files, align them to the mm10 reference genome, and perform filtering, barcode counting, and UMI counting. CellBender v0.3.0 (*Fleming et al., 2023*) was applied to remove counts from ambient/background RNA. The CellBender output files (.h5 format) were read into R (version 4.3.1) using Seurat 4.3.0, and each sample was further filtered prior to downstream analyses based on the number of UMIs (≥500) and genes (≥250), mitochondrial ratio (<15%), and the number of genes detected per UMI (>0.8). At the gene level, genes detected less than 10 nuclei and mitochondrial genes were filtered. To identify and remove doublets, we employed DoubletFinder v2.0 (*McGinnis et al., 2019*) using an estimated multiplet rate derived from the total number of recovered nuclei.

### Integration, clustering, annotation, and subclustering

Gene counts were first normalized using *SCTransform*, and regressed on mitochondrial read ratio and cell cycle score by Seurat. For data integration among samples, we performed canonical correlation analysis to identify shared sources of variation among samples, focusing on 3000 genes. Next, we employed the *FindIntegrationAnchors* function to determine mutual nearest neighbors, commonly referred to as anchors. Then, utilizing the identified anchors, the datasets were integrated using the *IntegrateData* function. For graph-based clustering, we visualized the integrated data using dimensionality reduction techniques, such as principal component analysis and uniform manifold approximation and projection, and clustered the nuclei with a resolution of 0.6. As a result, clusters were annotated as adipocytes, APC, mesothelial cells, endothelial cells, PC/SMC, macrophages, dendritic cells, mast cells, T cells, and B cells using their marker genes. For further investigation, the integrated data were subdivided into separate objects by cell type and re-clustered to achieve higher resolutions, with minimal additional filtering applied to potential doublets. To understand distinct functionality of subclusters, if needed pathway analysis was performed on the marker genes using clusterProfiler v4.8.2 (*Wu et al., 2021*). For visualization of gene expression levels within the specific cells, we used the R package called scCustomize (*Marsh, 2023*) which generates plots where cells expressing the gene of interest can be superimposed on top of the negative cells with the argument of 'order=TRUE'.

### Pseudobulk differential expression analysis and downstream analyses

To identify genes differentially expressed between chow and HFD conditions within particular cell types, we aggregated the counts and metadata to the sample level and compared the counts between conditions using edgeR v.3.42.4 (*Robinson et al., 2010*). Significance was determined based on false discovery rate <0.05 and absolute $\log_2$ fold change >1. Differentially expressed genes in either eWAT or iWAT were clustered using *K*-means clustering through Morpheus (https://software.broadinstitute.org/morpheus/), and each group of genes were passed to Metascape (*Zhou et al., 2019*) for pathway analysis.

### Single-cell trajectory analysis

We used Monocle 3 (*Cao et al., 2019*) to infer trajectories within cell types, potentially representing the sequences of gene expression changes that each cell/nucleus undergoes as part of a dynamic biological process. This analysis was conducted using the Seurat object containing subclustering information.

### Analysis of published data

For comparison of data quality, scRNA-seq or snRNA-seq datasets from previous studies were obtained from *Emont et al., 2022* (GSE176171) and *Sárvári et al., 2021* (GSE160729). The data were processed using Cell Ranger and read into R using Seurat. The median number of UMIs and genes detected per cell, the fraction of reads in cell, and the median number of mitochondrial read ratios were evaluated.

## Statistical analysis

Statistical details are available in the figure legends. Unless otherwise noted, all bar graphs and scatterplots show both mean ± SEM and individual values. p values were calculated using GraphPad Prism v10.0.0. and represent the results of Student's *t* test, one-way ANOVA with multiple comparisons, or linear regression test.

## Acknowledgements

We thank Dr. Hongyu Gao for providing guidance in setting up the computational analysis pipeline, and Dr. Charlie Dong for sharing equipment. We are grateful to the Flow Cytometry Core and the Center for Medical Genomics at Indiana University School of Medicine. This study was supported by National Institute of Diabetes and Digestive and Kidney Diseases (R01DK129289), and American Diabetes Association Junior Faculty Award (7-21-JDF-056) to H.C.R. This research was supported in part by Lilly Endowment, Inc, through its support for the Indiana University Pervasive Technology Institute.

## Additional information

### Competing interests

Gang Peng: Reviewing editor, eLife. The other authors declare that no competing interests exist.

### Funding

| Funder | Grant reference number | Author |
| --- | --- | --- |
| National Institute of Diabetes and Digestive and Kidney Diseases | R01DK129289 | Hyun Cheol Roh |
| American Diabetes Association | 7-21-JDF-056 | Hyun Cheol Roh |

The funders had no role in study design, data collection, and interpretation, or the decision to submit the work for publication.

### Author contributions

Jisun So, Data curation, Formal analysis, Investigation, Visualization, Methodology, Writing – original draft, Writing – review and editing; Olivia Strobel, Investigation; Jamie Wann, Kyungchan Kim, Avishek Paul, Methodology; Dominic J Acri, Quidance for computational analysis; Luke C Dabin, Quidance for computational analysis; Jungsu Kim, Quidance for computational analysis; Gang Peng, Guidance for computational analysis; Hyun Cheol Roh, Conceptualization, Resources, Data curation, Formal analysis, Supervision, Funding acquisition, Investigation, Methodology, Writing – original draft, Project administration, Writing – review and editing

### Author ORCIDs

Jisun So ⬤ https://orcid.org/0000-0001-8146-8139
Jungsu Kim ⬤ https://orcid.org/0000-0002-6931-8581
Hyun Cheol Roh ⬤ https://orcid.org/0000-0002-8176-5747

### Ethics

All animal experiments during this study were performed according to a protocol (#22073) approved by the Indiana University School of Medicine (IUSM) Institutional Animal Care and Use Committee (IACUC).

Reviewer #1 (Public review): https://doi.org/10.7554/eLife.97981.3.sa1
Reviewer #2 (Public review): https://doi.org/10.7554/eLife.97981.3.sa2
Reviewer #3 (Public review): https://doi.org/10.7554/eLife.97981.3.sa3
Author response https://doi.org/10.7554/eLife.97981.3.sa4

# Additional files

## Supplementary files

Supplementary file 1. Distinct biological pathways affected in adipocyte subpopulations by high fat diet (HFD). *K*-means clustering of differentially expressed genes (*n* = 4236) in at least one adipocyte subpopulation in either epididymal white adipose tissue (eWAT) or inguinal white adipose tissue (iWAT), based on their expression patterns across samples, results in four clusters. Each tab includes pathways enriched in each gene cluster, as determined by Metascape. Specific pathways highlighted in *Figure 5* are marked with borders.

Supplementary file 2. Sequences of primers used for quantitative real-time PCR (qRT-PCR).

Source code 1. An R script containing Seurat analysis for data integration, clustering, and annotation.

Source code 2. An R script containing adipocyte-specific analyses.

Source code 3. An R script containing adipocyte-specific pseudobulk differential expression analysis.

Source code 4. An R script containing adipocyte-specific pseudotime analysis using Monocle 3.

Source code 5. An R script containing analyses specific to cell types other than adipocytes.

Source code 6. An R script containing a comparative analysis to the previously published work by *Emont et al., 2022*.

MDAR checklist

## Data availability

Sequencing data have been deposited in the Gene Expression Omnibus (GEO) under accession codes GSE241987 (snRNA-seq) and GSE261417 (bulk nuclear RNA-seq). Data analysis codes used in this study are provided as Source code 1–6.

The following datasets were generated:

| Author(s) | Year | Dataset title | Dataset URL | Database and Identifier |
|---|---|---|---|---|
| Roh HC, So J, Paul A | 2024 | Robust snRNA-seq protocol revealing mouse white adipose tissue remodeling during obesity | https://www.ncbi.nlm.nih.gov/geo/query/acc.cgi?acc=GSE241987 | NCBI Gene Expression Omnibus, GSE241987 |
| Roh HC, So J | 2024 | Robust single nucleus RNA sequencing reveals depot-specific cell population dynamics in adipose tissue remodeling during obesity | https://www.ncbi.nlm.nih.gov/geo/query/acc.cgi?acc=GSE261417 | NCBI Gene Expression Omnibus, GSE261417 |

The following previously published datasets were used:

| Author(s) | Year | Dataset title | Dataset URL | Database and Identifier |
|---|---|---|---|---|
| Rosen ED, Tsai LT, Emont MP | 2022 | A single cell atlas of human adipose tissue | https://www.ncbi.nlm.nih.gov/geo/query/acc.cgi?acc=GSE176171 | NCBI Gene Expression Omnibus, GSE176171 |
| Sárvári AK, Van Hauwaert EL, Markussen LK, Gammelmark E, Marcher A, Ebbesen MF, Nielsen R, Brewer JR, Madsen JG, Mandrup S | 2021 | Plasticity of epididymal adipose tissue in response diet-induced obesity at single-nucleus resolution | https://www.ncbi.nlm.nih.gov/geo/query/acc.cgi?acc=GSE160729 | NCBI Gene Expression Omnibus, GSE160729 |

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
