## [Editor Report · eLife Assessment]

So et al. present an optimized protocol for single-nuclei RNA sequencing of adipose tissue in mice, ensuring better RNA quality and nuclei integrity. The authors use this protocol to explore the cellular landscape in both lean and diet-induced obese mice, identifying a dysfunctional hypertrophic adipocyte subpopulation linked to obesity. The data analyses are **solid**, and the findings are supported by the evidence presented. This study provides **valuable** information for the field of adipose tissue biology and will be particularly helpful for researchers using single-nuclei transcriptomics in various tissues.

---

## [Referee Report · Reviewer #1 (Public review)]

Summary:

This manuscript from So et al. describes what is suggested to be an improved protocol for single-nuclei RNA sequencing (snRNA-seq) of adipose tissue. The authors provide evidence that modifications to the existing protocols result in better RNA quality and nuclei integrity than previously observed, with ultimately greater coverage of the transcriptome upon sequencing. Using the modified protocol, the authors compare the cellular landscape of murine inguinal and perigonadal white adipose tissue (WAT) depots harvested from animals fed a standard chow diet (lean mice) or those fed a high-fat diet (mice with obesity).

Strengths:

Overall, the manuscript is well written, and the data are clearly presented. The strengths of the manuscript rest in the description of an improved protocol for snRNA-seq analysis. This should be valuable for the growing number of investigators in the field of adipose tissue biology that are utilizing snRNA-seq technology, as well as those other fields attempting similar experiments with tissues possessing high levels of RNAse activity.

Moreover, the study makes some notable observations that provide the foundation for future investigation. One observation is the correlation between nuclei size and cell size, allowing for the transcriptomes of relatively hypertrophic adipocytes in perigonadal WAT to be examined. Another notable observation is the identification of an adipocyte subcluster (Ad6) that appears "stressed" or dysfunctional and likely localizes to crown-like inflammatory structures where pro-inflammatory immune cells reside.

Weaknesses:

Analogous studies have been reported in the literature, including a notable study from Savari et al. (Cell Metabolism). This somewhat diminishes the novelty of some of the biological findings presented here. This is deemed a minor criticism as the primary goal is to provide a resource for the field.

---

## [Referee Report · Reviewer #2 (Public review)]

Summary:

In the present manuscript So et al describe an optimized method for nuclei isolation and single nucleus RNA sequencing (snRNA-Seq), which they use to characterize cell populations in lean and obese murine adipose tissues.

Strengths:

The detailed description of the protocol for single-nuclei isolation incorporating VRC may be useful to researchers studying adipose tissues, which contain high levels of RNAses.

While the majority of the findings largely confirm previous published adipose data sets, the authors present a detailed description of a mature adipocyte sub-cluster that appears to represent stressed or dying adipocytes present in obesity, and which is better characterized using the described protocol.

Weaknesses:

The use of VRC to enhance snRNA-seq has been previously published in other tissues, somewhat diminishing the novelty of this protocol.

The snRNA-seq data sets presented in this manuscript, when compared with numerous previously published single-cell analysis of adipose tissue, represent an incremental contribution. The nuclei-isolation protocol may represent an improvement in transcriptional analysis for mature adipocytes, however other stromal populations may be better sequenced using single intact-cell cytoplasmic RNA-Seq.

---

## [Referee Report · Reviewer #3 (Public review)]

The authors aimed to improve single-nucleus RNA sequencing (snRNA-seq) to address current limitations and challenges with nuclei and RNA isolation quality. They successfully developed a protocol that enhances RNA preservation and yields high-quality snRNA-seq data from multiple tissues, including a challenging model of adipose tissue. They then applied this method to eWAT and iWAT from mice fed either a normal or high-fat diet, exploring depot-specific cellular dynamics and gene expression changes during obesity. Their analysis included subclustering of SVF cells and revealed that obesity promotes a transition in APCs from an early to a committed state and induces a pro-inflammatory phenotype in immune cells, particularly in eWAT. In addition to SVF cells, they discovered six adipocyte subpopulations characterized by a gradient of unique gene expression signatures. Interestingly, a novel subpopulation, termed Ad6, comprised stressed and dying adipocytes with reduced transcriptional activity, primarily found in eWAT of mice on a high-fat diet. Overall, the methodology is sound, and the data presented supports the conclusions drawn. Further research based on these findings could pave the way for potential novel interventions in obesity and metabolic disorders, or for similar studies in other tissues or conditions.

Strengths:

The authors have presented a compelling set of results. They have compared their data with two previously published datasets and provide novel insight into the biological processes underlying mouse adipose tissue remodeling during obesity. The results are generally consistent and robust. The revised Discussion is comprehensive and puts the work in the context of the field.

Weaknesses:

• The adipose tissues were collected after 10 weeks of high-fat diet treatment, lacking the intermediate time points for identifying early markers or cell populations during the transition from healthy to pathological adipose tissue.

• The expansion of the Ad6 subpopulation in obese iWAT and gWAT is interesting. The author claims that Ad6 exhibited a substantial increase in eWAT and a moderate rise in iWAT (Figure 4C). However, this adipocyte subpopulation remains the most altered in iWAT upon obesity. Could the authors elaborate on why there is a scarcity of adipocytes with ROS reporter and B2M in obese iWAT?

• While the study provides extensive data on mouse models, the potential translation of these findings to human obesity remains uncertain.

Revised version: The authors have properly revised the paper in response to the above questions, and I have no other concerns.

---

## [Author Response]

The following is the authors’ response to the original reviews.

**Public Reviews:**

**Reviewer #1 (Public Review):**
Summary:This manuscript from So et al. describes what is suggested to be an improved protocol for single-nuclei RNA sequencing (snRNA-seq) of adipose tissue. The authors provide evidence that modifications to the existing protocols result in better RNA quality and nuclei integrity than previously observed, with ultimately greater coverage of the transcriptome upon sequencing. Using the modified protocol, the authors compare the cellular landscape of murine inguinal and perigonadal white adipose tissue (WAT) depots harvested from animals fed a standard chow diet (lean mice) or those fed a high-fat diet (mice with obesity).Strengths:Overall, the manuscript is well-written, and the data are clearly presented. The strengths of the manuscript rest in the description of an improved protocol for snRNA-seq analysis. This should be valuable for the growing number of investigators in the field of adipose tissue biology that are utilizing snRNA-seq technology, as well as those other fields attempting similar experiments with tissues possessing high levels of RNAse activity.Moreover, the study makes some notable observations that provide the foundation for future investigation. One observation is the correlation between nuclei size and cell size, allowing for the transcriptomes of relatively hypertrophic adipocytes in perigonadal WAT to be examined. Another notable observation is the identification of an adipocyte subcluster (Ad6) that appears "stressed" or dysfunctional and likely localizes to crown-like inflammatory structures where proinflammatory immune cells reside.Weaknesses:Analogous studies have been reported in the literature, including a notable study from Savari et al. (Cell Metabolism). This somewhat diminishes the novelty of some of the biological findings presented here. Moreover, a direct comparison of the transcriptomic data derived from the new vs. existing protocols (i.e. fully executed side by side) was not presented. As such, the true benefit of the protocol modifications cannot be fully understood.

We agree with the reviewer’s comment on the limitations of our study. Following the reviewer's suggestion, we performed a new analysis by integrating our data with those from the study by Emont et al. Please refer to the Recommendation for authors section below for further details.

**Reviewer #2 (Public Review):**
Summary:In the present manuscript So et al utilize single-nucleus RNA sequencing to characterize cell populations in lean and obese adipose tissues.Strengths:The authors utilize a modified nuclear isolation protocol incorporating VRC that results in higherquality sequencing reads compared with previous studies.Weaknesses:The use of VRC to enhance snRNA-seq has been previously published in other tissues. The snRNA-seq snRNA-seq data sets presented in this manuscript, when compared with numerous previously published single-cell analyses of adipose tissue, do not represent a significant scientific advance.Figure 1-3: The snRNA-seq data obtained by the authors using their enhanced protocol does not represent a significant improvement in cell profiling for the majority of the highlighted cell types including APCs, macrophages, and lymphocytes. These cell populations have been extensively characterized by cytoplasmic scRNA-seq which can achieve sufficient sequencing depth, and thus this study does not contribute meaningful additional insight into these cell types. The authors note an increase in the number of rare endothelial cell types recovered, however this is not translated into any kind of functional analysis of these populations.

We acknowledge the reviewer's comments on the limitations of our study, particularly the lack of extension of our snRNA-seq data into functional studies of new biological processes. However, this manuscript has been submitted as a Tools and Resources article. As an article of this type, we provide detailed information on our snRNA-seq methods and present a valuable resource of high-quality mouse adipose tissue snRNA-seq data. In addition, we demonstrate that our improved method offers novel biological insights, including the identification of subpopulations of adipocytes categorized by size and functionality. We believe this study offers powerful tools and significant value to the research community.

Figure 4: The authors did not provide any evidence that the relative fluorescent brightness of GFP and mCherry is a direct measure of the nuclear size, and the nuclear size is only a moderate correlation with the cell size. Thus sorting the nuclei based on GFP/mCherry brightness is not a great proxy for adipocyte diameter. Furthermore, no meaningful insights are provided about the functional significance of the reported transcriptional differences between small and large adipocyte nuclei.

To address the reviewer's point, we analyzed the Pearson correlation coefficient for nucleus size vs. adipocyte size and found R = 0.85, indicating a strong positive correlation. In addition, we performed a new experiment to determine the correlation between nuclear GFP intensity and adipocyte nucleus size, finding a strong correlation with R = 0.91. These results suggest that nuclear GFP intensity can be a strong proxy for adipocyte size. Furthermore, we performed gene ontology analysis on genes differentially regulated between large and small adipocyte nuclei. We found that large adipocytes promote processes involved in insulin response, vascularization and DNA repair, while inhibiting processes related to cell migration, metabolism and the cytoskeleton. We have added these new data as Figure 4E, Figure 4-figure supplement 1E, G, and H (page 11)

Figure 5-6: The Ad6 population is highly transcriptionally analogous to the mAd3 population from Emont et al, and is thus not a novel finding. Furthermore, in the present data set, the authors conclude that Ad6 are likely stressed/dying hypertrophic adipocytes with a global loss of gene expression, which is a well-documented finding in eWAT > iWAT, for which the snRNA-seq reported in the present manuscript does not provide any novel scientific insight.

As the reviewer pointed out, a new analysis integrating our data with the previous study found that Ad3 from our study is comparable to mAd3 from Emont et al. in gene expression profiles. However, significant discrepancies in population size and changes in response to obesity were observed, likely due to differences in technical robustness. The dysfunctional cellular state of this population, with compromised RNA content, may have hindered accurate capture in the previous study, while our protocol enabled precise detection. This underscores the importance of our improved snRNA-seq protocol for accurately understanding adipocyte population dynamics. We have revised the manuscript to include new data in Figure 6-figure supplement 1 (page 14).

**Reviewer #3 (Public Review):**
Summary:The authors aimed to improve single-nucleus RNA sequencing (snRNA-seq) to address current limitations and challenges with nuclei and RNA isolation quality. They successfully developed a protocol that enhances RNA preservation and yields high-quality snRNA-seq data from multiple tissues, including a challenging model of adipose tissue. They then applied this method to eWAT and iWAT from mice fed either a normal or high-fat diet, exploring depot-specific cellular dynamics and gene expression changes during obesity. Their analysis included subclustering of SVF cells and revealed that obesity promotes a transition in APCs from an early to a committed state and induces a pro-inflammatory phenotype in immune cells, particularly in eWAT. In addition to SVF cells, they discovered six adipocyte subpopulations characterized by a gradient of unique gene expression signatures. Interestingly, a novel subpopulation, termed Ad6, comprised stressed and dying adipocytes with reduced transcriptional activity, primarily found in eWAT of mice on a high-fat diet. Overall, the methodology is sound, the writing is clear, and the conclusions drawn are supported by the data presented. Further research based on these findings could pave the way for potential novel interventions in obesity and metabolic disorders, or for similar studies in other tissues or conditions.Strengths:• The authors developed a robust snRNA-seq technique that preserves the integrity of the nucleus and RNA across various tissue types, overcoming the challenges of existing methods.• They identified adipocyte subpopulations that follow adaptive or pathological trajectories during obesity.• The study reveals depot-specific differences in adipose tissues, which could have implications for targeted therapies.Weaknesses:• The adipose tissues were collected after 10 weeks of high-fat diet treatment, lacking the intermediate time points for identifying early markers or cell populations during the transition from healthy to pathological adipose tissue.

We agree with the reviewers regarding the limitations of our study. To address the reviewer’s comment, we revised the manuscript to include this in the Discussion section (page 17).

• The expansion of the Ad6 subpopulation in obese iWAT and gWAT is interesting. The author claims that Ad6 exhibited a substantial increase in eWAT and a moderate rise in iWAT (Figure 4C). However, this adipocyte subpopulation remains the most altered in iWAT upon obesity. Could the authors elaborate on why there is a scarcity of adipocytes with ROS reporter and B2M in obese iWAT?

We observed an increase in the levels of H2DCFA reporter and B2M protein fluorescence in adipocytes from iWAT of HFD-fed mice, although this increase was much less compared to eWAT, as shown in Figure 6B (left panel). These increases in iWAT were not sufficient for most cells to exceed the cutoff values used to determine H2DCFA and B2M positivity in adipocytes during quantitative analysis. We have revised the manuscript to clarify these results (page 13).

• While the study provides extensive data on mouse models, the potential translation of these findings to human obesity remains uncertain.

To address the reviewer’s point, we expanded our discussion on the differences in adipocyte heterogeneity between mice and humans. We attempted to identify human adipocyte subclusters that resemble the metabolically unhealthy Ad6 adipocytes found in mice in our study; however, we did not find any similar adipocyte types. It has been reported that human adipocyte heterogeneity does not correspond well to that of mouse adipocytes (Emont et al. 2022). In addition, the heterogeneity of human adipocyte populations is not reproducible between different studies (Massier et al. 2023). Interestingly, this inconsistency is unique to adipocytes, as other cell types in adipose tissues display reproducible sub cell types across species and studies (Massier et al. 2023). Our findings indicate that adipocytes may exhibit a unique pathological cellular state with significantly reduced RNA content, which may contribute to the poor consistency in adipocyte heterogeneity in prior studies with suboptimal RNA quality. Therefore, using a robust method to effectively preserve RNA quality may be critical for accurately characterizing adipocyte populations, especially in disease states. It may be important to test in future studies whether our snRNA-seq protocol can identify consistent heterogeneity in adipocyte populations across different species, studies, and individual human subjects. We have revised the manuscript to include this new discussion (page 17).

**Recommendations for the authors:**

**Reviewer #1 (Recommendations For The Authors):**
Suggested points to address:(1) The authors suggest that their improved protocol for maintaining RNA/nucleus integrity results in a more comprehensive analysis of adipose tissue heterogeneity. The authors compare the quality of their snRNA-seq data to those generated in prior studies (e.g., Savari et al.). What is not clear is whether additional heterogeneity/clusters can be observed due directly to the protocol modifications. A direct head-to-head comparison of the protocols executed in parallel would of course be ideal; however, integrating their new dataset with the corresponding data from Savari et al. could help address this question and help readers understand the benefits of this new protocol vs. existing protocols.

The data from Savari et al. are of significantly lower quality, likely because they were generated using earlier versions of the 10X Genomics system, and this study lacks iWAT data. To address the reviewer’s point, we instead integrated our data with those from the other study by Emont et al. (2022), which used comparable tissue types and experimental systems. The integrated analysis confirmed the improved representation of all cell types present in adipose tissues in our study, with higher quality metrics such as increased Unique Molecular Identifiers (UMIs) and the number of genes per nucleus. These results indicate that our protocol offers significant advantages in generating a more accurate representation of each cell type and their gene expression profiles. New data are included in Figure 2-figure supplement 2 (page 7).

(2) The exact frequency of the Ad6 population in eWAT of mice maintained on HFD is a little unclear. From the snRNA-seq data, it appears that roughly 47% of the adipocytes are in this "stressed state." In Figure 6, it appears that greater than 75% of the adipocytes express B2M (Ad6 marker) and greater than 75% of adipocytes are suggested to be devoid of measurable PPARg expression. The latter seems quite high as PPARg expression is essential to maintain the adipocyte phenotype. Is there evidence of de-differentiation amongst them (i.e. acquisition of progenitor cell markers)? Presenting separate UMAPs for the chow vs. HFD state may help visualize the frequency of each adipocyte population in the two states. Inclusion of the stromal/progenitor cells in the visualization may help understand if cells are de-differentiating in obesity as previously postulated by the authors. Related to Point # 1 above, is this population observed in prior studies and at a similar frequency?

To address the reviewer’s point, we analyzed the expression of adipocyte progenitor cell (APC) markers, such as Pdgfra, in the Ad6 population. We did not detect significant expression of APC markers, suggesting that Ad6 does not represent dedifferentiating adipocytes. Instead, they are likely stressed and dying cells characterized by an aberrant state of transcription with a global decline.

When integrating our data with the datasets by Emont et al., we observed an adipocyte population in the previous study, mAd3, comparable to Ad6 in our study, with similar marker gene expression and lower transcript abundance. However, the population size of mAd3 was much smaller than that of Ad6 in our data and did not show consistent population changes during obesity. This discrepancy may be due to different technical robustness; the dysfunctional cellular state of this population, with its severely compromised RNA contents, may have made it difficult to accurately capture using standard protocols in the previous study, while our protocol enabled robust and precise detection. We added new data in Figure 4-figure supplement 1I and Figure 6-figure supplement 17 (page 14) and revised the Discussion (page 17).

Additional points(1) The authors should be cautious in describing subpopulations as "increasing" or "decreasing" in obesity as the data are presented as proportions of a parent population. A given cell population may be "relatively increased."

To address the reviewer's point, we revised the manuscript to clarify the "relative" changes in cell populations during obesity in the relevant sections (pages 8, 9, 10, 11, and 15).

(2) The authors should also be cautious in ascribing "function" to adipocyte populations based solely on their expression signatures. Statements such as those in the abstract, "...providing novel insights into the mechanisms orchestrating adipose tissue remodeling during obesity..." should probably be toned down as no such mechanism is truly demonstrated.

To address the reviewer's point, we revised the manuscript by removing or replacing the indicated terms or phrases with more suitable wording in the appropriate sections (page 2, 10, 12, 14)

**Reviewer #3 (Recommendations For The Authors):**
(1) The authors might consider expanding a discussion on the potential implications of their findings, especially the newly identified adipocyte subpopulations and depot-specific differences for human studies.

To address the reviewer’s point, we attempted to identify human adipocyte subclusters that resembled our dysfunctional Ad6 adipocytes in mice; however, we did not find any similar adipocyte types. It has been reported that human adipocyte heterogeneity does not correspond well to that of mouse adipocytes (Emont et al. 2022). In addition, the heterogeneity of human adipocyte populations is not reproducible between different studies (Massier et al. 2023). Interestingly, this inconsistency is unique to adipocytes, as other cell types in adipose tissues display reproducible sub cell types across species and studies (Massier et al. 2023). Our findings indicate that adipocytes may exhibit a unique pathological cellular state with significantly reduced RNA content, which may contribute to the poor consistency in adipocyte heterogeneity in prior studies with suboptimal RNA quality. Therefore, using a robust method to effectively preserve RNA quality may be critical for accurately characterizing adipocyte populations, especially in disease states. It may be important to test in future studies whether our snRNA-seq protocol can identify consistent heterogeneity in adipocyte populations across different species, studies, and individual human subjects. We have revised the manuscript to include this new discussion (page 17)

(2) typo: "To generate diet-induced obesity models".

We revised the manuscript to correct it.